# GRepsNet: A Simple Equivariant Network for Arbitrary Matrix Groups

## Abstract

Group equivariance is a strong inductive bias useful in a wide range of domains including images, point clouds, dynamical systems, and partial differential equations (PDEs). But constructing efficient equivariant networks for general groups and domains is difficult. Recent work by Finzi et al. (2021) directly solves the equivariance constraint for arbitrary matrix groups to obtain equivariant MLPs (EMLPs). However, this method does not scale well and scaling is crucial to get the best from deep learning. This necessitates the design of group equivariant networks for general domains and groups that are simple and scalable. To this end, we introduce Group Representation Networks (GRepsNets), a simple equivariant network for arbitrary matrix groups. The key intuition for our design is that using tensor representations in the hidden layers of a neural network along with appropriate *mixing* of various representations can lead to expressive equivariant networks, which we confirm empirically. We find GRepsNet to be competitive to EMLP on several tasks with group symmetries such as O(5), O(1, 3), and O(3) with scalars, vectors, and second-order tensors as data types. To illustrate the simplicity and generality of our network, we also use it for image classification with MLP-mixers, predicting N-body dynamics using message passing neural networks (MPNNs), and for solving PDEs using Fourier neural operators (FNOs). Surprisingly, we find that using simple first-order representations itself can yield benefits of group equivariance without additional changes in the architecture. Finally, we illustrate how higher-order tensor representations can be used for group equivariant finetuning that outperforms the existing equivariant finetuning method Basu et al. (2023b).

## 1 Introduction

Group equivariance plays a key role in the success of several popular architectures such as translation equivariance in Convolutional Neural Networks (CNNs) for image processing (LeCun et al., 1989), 3D rotational equivariance in Alphafold2 (Jumper et al., 2021), and equivariance to general discrete groups in Group Convolutional Neural Networks (GCNNs) (Cohen & Welling, 2016a).

But designing efficient equivariant networks can be challenging both because they require domain-specific knowledge and can be computationally inefficient. E.g., there are several works designing architectures for different groups such as the special Euclidean group SE(3) (Fuchs et al., 2020), special Lorentz group O(1, 3) (Bogatskiy et al., 2020), discrete Euclidean groups (Cohen & Welling, 2016a; Ravanbakhsh et al., 2017), etc. Moreover, some of these networks can be computationally inefficient, prompting the design of simpler and lightweight equivariant networks such as E(n) equivariant graph neural networks (Satorras et al., 2021) for graphs and vector neurons (Deng et al., 2021) for point cloud processing.

Finzi et al. (2021) propose an algorithm to construct equivariant MLPs (EMLPs) for arbitrary matrix groups when the data is provided using tensor polynomial representations. This method directly computes the basis of the equivariant MLPs and requires minimal domain knowledge. Although elegant, EMLPs are restricted to MLPs or can be used as subcomponents in larger networks, and are not useful for making more general architectures equivariant as a whole. Moreover, using equivariant basis functions can often be computationally expensive (Fuchs et al., 2020; Thomas et al., 2018) leading to several group-specific efficient architectures. Equivariance as an inductive bias makes the learning problem easier and provides robustness guarantees, and scaling is important to learn more

complex functions that is not easy to model using inductive biases such as equivariance. So, we need an architecture that is equivariant and allows scaling to larger sizes and datasets.

To this end, we introduce Group Representation Network (GRepsNet), which replaces scalar representation from classical neural networks with tensor representations of different orders to obtain expressive equivariant networks. This is reminiscent of vector neurons that introduce SO(3) representations in various architectures to obtain equivariance to the SO(3) group. In contrast, GRepsNet works for arbitrary matrix groups like EMLPs and can also leverage higher-order tensor representations, unlike vector neurons.

We perform three sets of experiments: a) on synthetic datasets from Finzi et al. (2021) to provide a direct comparison with EMLP across different groups and tensor representations; b) on image classification using MLP-Mixers (Tolstikhin et al., 2021), on N-body dynamics dataset using message passing neural networks (MPNNs) (Gilmer et al., 2017), and solving PDEs using Fourier Neural Operators (FNOs) (Li et al., 2021; Helwig et al., 2023) to show the simplicity and generality of our proposed method across a range of data and architectures; c) showing the use of higher-order tensor representations in equivariant finetuning from pretrained models (Basu et al., 2023b). Our main contributions are summarized below.

1. We propose a simple equivariant architecture called GRepsNet, equivariant to arbitrary matrix groups that perform competitively with EMLP on several groups such as O(5), O(3), and O(1, 3) using scalars, vectors, and second-order tensor representations.

2. We find that using GRepsNet with simple representations gives competitive results with several architectures used in different domains such as image classification, PDEs, and N-body dynamics predictions using MLP-mixers, FNOs, and MPNNs, respectively.

3. We leverage second-order tensor features for equivariant image classification using CNNs. When used for finetuning, it outperforms equituning (Basu et al., 2023b) that uses first-order representations.

## 2 RELATED WORKS

**Parameter sharing** A popular method for constructing group equivariant architectures involves sharing learnable parameters in the network to guarantee equivariance, e.g. CNNs (LeCun et al., 1989), GCNNs (Cohen & Welling, 2016a; Kondor & Trivedi, 2018), Deepsets (Zaheer et al., 2017), etc. However, all these methods are restricted to discrete groups, unlike our work which can handle equivariance to arbitrary matrix groups.

**Steerable networks** Another popular approach for constructing group equivariant networks is by first computing a basis of the space of equivariant functions, then linearly combining these basis vectors to construct an equivariant network. This method can also handle continuous groups. Several popular architectures employ this method, e.g. steerable CNNs (Cohen & Welling, 2016b), E(2)-CNNs (Weiler & Cesa, 2019), Tensor Field Networks (Thomas et al., 2018), SE(3)-transformers (Fuchs et al., 2020), EMLPs Finzi et al. (2021) etc. But, these methods are computationally expensive, thus, often replaced by efficient equivariant architectures for specific models, e.g., E($n$) equivariant graph neural networks (Satorras et al., 2021) for graphs and vector neurons (Deng et al., 2021) for point cloud processing. More comparisons with EMLPs are provided in Sec. B.

**Representation-based methods** A simple alternative to using steerable networks for continuous networks is to construct equivariant networks by simply representing the data using group representations, only using scalar weights to combine these representations, and using non-linearities that respect their equivariance. Works that use representation-based methods include vector neurons (Deng et al., 2021) for O(3) group and universal scalars Villar et al. (2021). Vector neurons are restricted to first-order tensors and universal scalars face scaling issues, hence, mostly restricted to synthetic experiments. More comparisons with universal scalars are provided in Sec. B.

**Frame averaging** Yet another approach to obtain group equivariance is to use frame-averaging (Yarotsky, 2022; Puny et al., 2021), where averaging over equivariant frames corresponding to each input is performed to obtain equivariant outputs. This method works for both discrete and continuous groups but requires the construction of these frames, either fixed by design as in Puny et al. (2021); Basu et al. (2023b) or learned using auxiliary equivariant neural networks as in Kaba et al. (2023).

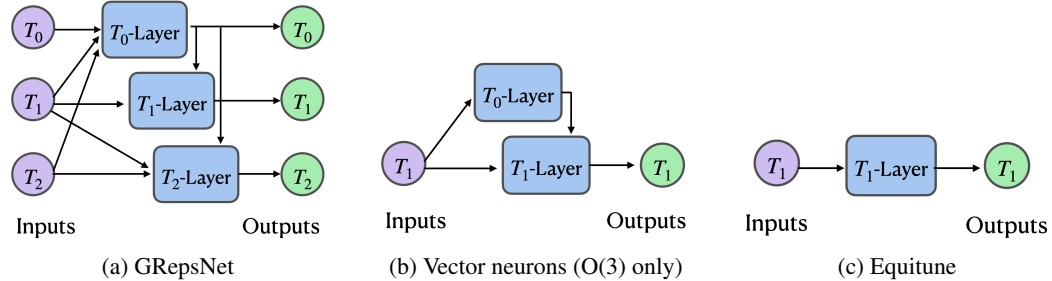

Figure 1: (a) An example of a GRepsNet layer with inputs of types $T_0, T_1$, and $T_2$, and outputs of the same types. The $T_i$ layer first converts all the inputs to type $T_i$ as described in §. 4.1 following which it is processed by a neural network layer, such as an MLP. (b) and (c) show layers from vector neurons Deng et al. (2021) and equitune Basu et al. (2023b), which are special cases of GRepsNet.

Our method is in general different from this approach since our method does not involve averaging over any frame or the use of auxiliary equivariant networks. For the special case of discrete groups, the notion of frame averaging is closely related to both parameter sharing as well as representation methods. Hence, in the context of equituning (Basu et al., 2023b), we show how higher-order tensor representations can directly be incorporated into their frame-averaging method.

## 3 BACKGROUND

### 3.1 GROUP AND REPRESENTATION THEORY

Basics of groups and group actions are provided in §. A. Let $GL(m)$ represent the group of all invertible matrices of dimension $m$. Then, for a group $G$, the **linear group representation** of $G$ is defined as the map $\rho : G \mapsto GL(m)$ such that $\rho(g_1 g_2) = \rho(g_1)\rho(g_2)$ and $\rho(e) = I$, the identity matrix. A group representation of dimension $m$ is a linear group action on the vector space $\mathbb{R}^m$.

Given some base linear group representation $\rho(g)$ for $g \in G$ on some vector space $V$, we construct **tensor representations** by applying Kronecker sum $\oplus$, Kronecker product $\otimes$, and tensor dual $*$. Each of these tensor operations on the vector spaces lead to corresponding new group actions. The group action corresponding to $V^*$ becomes $\rho(g^{-1})^T$. Let $\rho_1(g)$ and $\rho_2(g)$ for $g \in G$ be group actions on vector spaces $V_1$ and $V_2$, respectively. Then, the group action on $V_1 \oplus V_2$ is given by $\rho_1(g) \oplus \rho_2(g)$ and that on $V_1 \otimes V_2$ is given by $\rho_1(g) \otimes \rho_2(g)$.

We denote the tensors corresponding to the base representation $\rho$ as $T_1$ tensors, i.e., tensors of order one, and $T_0$ denotes a scalar. In general, $T_m$ denotes a tensor of order $m$. Further, Kronecker product of tensors $T_m$ and $T_n$ gives a tensor $T_{m+n}$ of order $m + n$. We use the notation $T_m^{\otimes r}$ to denote $r$ times Kronecker product of $T_m$ tensors. Kronecker sum of two tensors of types $T_m$ and $T_n$ gives a tensor of type $T_m \oplus T_n$. Finally, Kronecker sum of $r$ tensors of the same type $T_m$ is written as $rT_m$.

## 4 METHOD

### 4.1 GENERAL ARCHITECTURE

We describe the construction of the GRepsNet layer, an example of which shown in Fig. 1a. The GRepsNet model is then constructed by stacking several of the GRepsNet layers. Let the input to a GRepsNet layer be of type $\oplus_{i \in N} a_i T_i$, where $a_i$s are scalars indicating that the input has $a_i$ tensors of type $T_i$.

Each GRepsNet layer further has several $T_i$-layers as shown in Fig. 1a. Each $T_i$ layer performs two operations: a) converting the input tensors to appropriate tensor types, b) process the converted tensors using a neural network layer, such as an MLP or a CNN. If the representations used are not regular representations, we make the assumption that the input to the GRepsNet model always consist of some tensors with $T_1$ representations, which is not a strong assumption that helps keep our construction simple and also encompasses all experiments from Finzi et al. (2021).

**Convert tensor types** The input to a $T_i$-layer converts all inputs of types $T_j$, $j \neq i$ along with $T_1, T_0$ (when available) to type $T_i$ before passing it through an appropriate neural network layer, such as an MLP or a CNN.

We only convert the input type $T_j$ to $T_i$ when $i > j > 0$ or $i = 0$. Otherwise, we do not use $T_j$ in the $T_i$-layer. When $i > j > 0$, we first write $i$ as $i = kj + r$, where $k \in \mathrm{N}$ and $r < j$. We obtain the $T_i$ tensor using $T_j, T_1$, and $T_0$ as $T_j^{\otimes k} \otimes T_1^{\otimes r}$.

When $i = 0$, we convert each input of type $T_j$ to type $T_0$ by using an appropriate invariant operator, e.g. Euclidean norm for Euclidean groups. These design choices keeps our design lightweight as well as expressive as confirmed empirically. Details on how these inputs are processed is described next.

**Process converted tensors** When *not using regular representations*, first, the $T_0$-layer passes all the tensors of type $T_0$ or scalars through a neural network such as an MLP or a CNN. Since the inputs are invariant scalars, hence, the outputs are always invariant and thus, there are no restrictions on the neural network used for the $T_0$-layer, i.e., they may also use non-linearities.

Lets call the output from the $T_0$-layer as $Y_{T_0}$. For a $T_i$-layer with $i > 0$, we simply pass it through a linear neural network with no point-wise non-linearities or bias terms to ensure that the output is equivariant. Lets call this output $H_{T_i}$. Then, to mix the $T_i$ tensors with the $T_0$ tensors better, we update $H_{T_i}$ as $H_{T_i} = H_{T_i} * \frac{Y_{T_0}}{\mathrm{inv}(H_{T_i})}$, where $\mathrm{inv}(\cdot)$ is simply an invariant function such as the Euclidean norm for an Euclidean group. Finally, we pass $H_{T_i}$ through another linear layer without any bias or pointwise non-linearities to obtain $Y_{T_i}$.

When using *regular representations* for a discrete group of size $|G|$, the neural networks used also contain pointwise non-linearities and biases, as they do not affect the equivariance for regular representation. The proof of equivariance of our architecture is trivial and follows the proof of equivariance for vector neurons as provided in §. C. While the architecture is already simplified to make it easy to use, we further make more simplifications as needed for specific applications.

## 4.2 SPECIAL CASES AND RELATED DESIGNS

Here, we look at existing group equivariant architectures popular for their simplicity that are special cases or closely related to our general design.

**Vector neurons** Popular for its lightweight $SO(3)$-equivariant applications such as point cloud, the vector neurons (Deng et al., 2021) serve as a classic example of special cases of our design as illustrated in Fig. 1b. Their $T_1$-layer simply consists of a linear combination $T_1$ inputs without bias terms, same as ours. The $T_0$-layer first converts the $T_1$ tensors into $T_0$ tensors by taking inner products. Then, pointwise non-linearities are applied to the $T_0$ tensor and then mixed with the $T_1$ tensors, by multiplying them with $T_1$ tensors and further linearly mixing the $T_1$ tensors.

**Harmonic networks** Harmonic networks or H-nets Worrall et al. (2017) employ a similar architecture to ours and vector neurons, but specialized for the $SO(2)$ group. They also take as input $T_1$ inputs, then obtain the $T_0$ scalars by computing the Euclidean norms of the inputs. All non-linearities are applied only to the scalars. The $T_1$ tensors are processed using linear circular cross-correlations that preserve equivariance. Further, higher order tensors are obtained by chained-cross correlations. The use of cross-correlations differ slightly from our design and that of vector neurons. But it is designed in a similar spirit of building tensors of various orders and construct simple, yet expressive equivariant features.

**Equitune** Finally, recent works on frame-averaging such as equitune, $\lambda$-equitune Basu et al. (2023a) and probabilistic symmetrization Kim et al. (2023) construct equivariant architecture by performing some sort of averaging over groups. This can be seen as using a regular $T_1$ representation as the input and output type as illustrated in Fig. 1c. These works have mainly focused on exploring the potential of equivariance in pretrained models. In this work, we further explore the capabilities of regular $T_1$ representations and find their surprising benefits in equivariant tasks. Moreover, this also inspires us to explore beyond regular $T_1$ representations, e.g., we find $T_2$ representations can yield better results than $T_1$ representations when used in the final layers of a model for image classification.

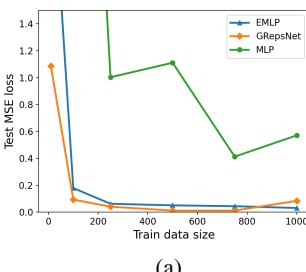 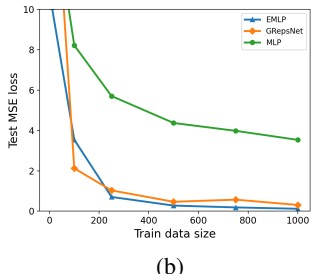 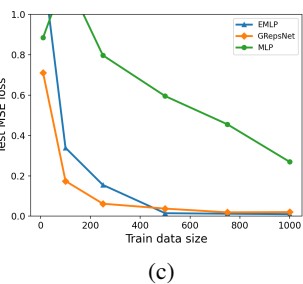

(a)        (b)        (c)

Figure 2: Comparison of GRepsNets with EMLPs Finzi et al. (2021) and MLPs for (a) O(5)-invariant synthetic regression task with input type $2T_1$ and output type $T_0$, (b) O(3)-equivariant regression with input as masses and positions of 5 point masses using representation of type $5T_0 + 5T_1$ and output as the inertia matrix of type $T_2$, (c) SO(1, 3)-invariant regression computing the matrix element in electron-muon particle scattering with input of type $4T_1$ and output of type $T_0$. Across all the tasks, we find that GRepsNets, despite its simple design, are competitive with the more sophisticated EMLPs and significantly outperform MLPs.

## 4.3 APPLICATIONS AND NETWORK DETAILS

Now, we provide details of applications considered and precise network designs used for each of them. An overview of the practical designs insights is provided in Sec. D.

**Synthetic experiments related to EMLP (Finzi et al., 2021)** We work with three group equivariant regression tasks for the O(5)-invariant regression, O(3)-equivariant regression, and O(1, 3)-invariant regression. The (input types, output types) corresponding to these tasks are $(2T_1, T_0)$, $(5T_0 + 5T_1, T_2)$, and $(4T_1, T_0)$. We use architectures with the given input and output types following the GRepsNet design in §. 4.1. As described in §. 4.1, once the tensor types are converted, we process the tensors using some neural networks. Here we use simple MLPs for this processing task.

**Image classification with MLP-mixers** Here we work with CIFAR10 (Krizhevsky et al.) with random 90° rotations, which we call rot90-CIFAR10, along with Galaxy10 (Leung & Bovy) and EuroSAT (Helber et al., 2019) datasets that have naturally 90° symmetries in them. We convert the images to $T_1$ tensors of the group $C_4$ of 90° rotations efficiently as described in §. D.2, which makes the design more efficient than the traditional regular representations, e.g., used in equitune by repeating transformed images in the input. Once the regular $T_1$ representations are obtained, we obtain an additional *group* dimension in the data in addition to the batch, channel, and spatial dimensions. The group dimension is treated like the batch dimension, exactly as done in equitune with $T_1$ regular representations.

**PDE solving with FNOs** We work with two versions of the incompressible Navier-Stokes equation from Helwig et al. (2023); Li et al. (2021), with and without symmetry with respect to 90° rotations. Here we use the $T_1$ representation for the $C_4$ group of 90° rotations exactly the way used in equitune. This is because, unlike image classification, here we work in the Fourier domain and it is crucial to preserve all the frequency modes of the original input. The precise way of construction of this representation is provided in §. D.3.

The group dimension is treated like the batch dimension similar to the case of image classification. Then, the data is sequentially passed through four FNO layers taken from Helwig et al. (2023). The ability to directly use various models such as FNOs and still preserving equivariance emphasizes the simplicity of our method.

**Predicting N-body dynamics using GNNs** We consider the problem of predicting dynamics of $N$ charged particles given their charges and initial positions, where the group of symmetry considered is the orthogonal group $O(3)$. Each particle is placed at a node of a graph $\mathcal{G} = \{\mathcal{V}, \mathcal{E}\}$, where $\mathcal{V}$ and $\mathcal{E}$ are the sets of vertices and edges. Let the edge attributes of $\mathcal{G}$ be $a_{ij}$, and let $h_i^l$ be the node feature of node $v_i \in \mathcal{V}$ at layer $l$ of a message passing neural network (MPNN). An MPNN

Table 1: We find simple GRepsMLP-Mixers with $T_1$ representations can leverage the benefits of equivariance with a very simple architecture design. Mean (std) of test accuracies over 3 seeds. Both GRepsMLP-Mixer-1 and GRepsMLP-Mixer-2 use regular $T_1$ representations, with architecture near-identical to the MLP-Mixer used. GRepsMLP-Mixer-2 uses a simple early fusion module from §. D.2 in addition to the late fusion present in GRepsMLP-Mixer-1. Our models clearly outperform non-equivariant MLP-Mixers with early fusion outperforming or performing competitively to late fusion for image classification tasks.

| Dataset \ Model | MLP-Mixer | GRepsMLP-Mixer-1 (ours) | GRepsMLP-Mixer-2 (ours) |
|---|---|---|---|
| Rot90-CIFAR10 | 75.08 (0.8) | 77.95 (0.37) | **79.09 (0.3)** |
| Galaxy 10 | 82.62 (0.3) | 85.02 (1.0) | **88.86 (0.2)** |
| Eurosat | 96.44 (0.1) | **97.83 (0.2)** | 97.28 (0.3) |

as defined by Gilmer et al. (2017) has an edge update, $m_{ij} = \phi_e(h_i^l, h_j^l, a_{ij})$ and a node update $h_i^{l+1} = \phi_h(h_i^l, m_i)$, $m_i = \sum_{j \in \mathcal{N}(i)} m_{ij}$, where $\phi_e$ and $\phi_h$ are MLPs corresponding to edge and node updates, respectively.

We design GRepsGNN by making small modifications to the MPNN architecture. In our model, we use two edge updates for $T_0$ and $T_1$ tensors, respectively, and one node update for $T_1$ update. The two edge updates are $m_{ij,T_0} = \phi_{e,T_0}(\|h_i^l\|, \|h_j^l\|, a_{ij})$, $m_{ij,T_1} = \phi_{e,T_1}(h_i^l, h_j^l, a_{ij})$, where $\|\cdot\|$ obtains $T_0$ tensors from $T_1$ tensors for the Euclidean group, $\phi_{e,T_0}(\cdot)$ is $T_0$-layer MLP, and $\phi_{e,T_1}(\cdot)$ is a $T_1$-layer made of an MLP without any pointwise non-linearities or biases. The final edge update is obtained as $m_{ij} = m_{ij,T_1} * m_{ij,T_0} / \|m_{ij,T_1}\|$. Finally, the node update is given by $h_i^{l+1} = \phi_{h,T_1}(h_i^l, m_i)$, where $m_i = \sum_{j \in \mathcal{N}(i)} m_{ij}$ and $\phi_{h,T_1}(\cdot)$ is an MLP without any pointwise non-linearities or biases. Thus, the final node update is a $T_1$ tensor.

**Equivariant image classification using CNNs with $T_2$ representations** Here we test the hypothesis that $T_2$ representation of features extracted by CNNs serves as better equivariant features than $T_1$ representations such as used in equitune Basu et al. (2023b). The group of symmetry considered here is the $\mathrm{C}N$ group of $\frac{360}{N}^\circ$ rotations. Our intuition arises from the fact that $T_2$ representations have better mixing amongst features in the group dimension than $T_1$ representations. This is because, $T_2$ representation stems from the outer product of two $T_1$ representations. This is also similar to the use of outer products in the features by Bilinear CNNs (Lin et al., 2015) leading to efficient processing of features for fine-grained classification. Our approach differs from Lin et al. (2015) in that our outer product is in the group dimension that preserves equivariance, whereas Lin et al. (2015) are not explicitly concerned with group equivariance.

We first use rot90-CIFAR10 to test our hypothesis that $T_2$ representation provide better performance than $T_1$ representations. We use a CNN with 3 convolutional layers followed by 5 linear layers. The initial $k$ layers of the network use $T_1$ representation, following which all layers use $T_2$ representations. We verify that best performance is obtained when the last few layers use $T_2$ representations.

Based on this observation, we propose $T_2$-equitune, that extracts $T_1$ features from pretrained models like Basu et al. (2023b), but converts them to $T_2$ representations before providing invariant outputs. We work with pretrained Resnet18 and finetune on rot90-CIFAR10 and Galaxy10 datasets. We corroborate our hypothesis and find that $T_2$-equitune outperforms equitune by simply using $T_2$ representations in the extracted features. We further provide comparison with GCNNs Cohen & Welling (2016a) and E(2)-CNNs Weiler & Cesa (2019).

# 5 DATASETS AND EXPERIMENTS

## 5.1 COMPARISON WITH EMLPS

**Datasets:** We consider three regression tasks from Finzi et al. (2021): O(5)-invariant task, O(3)-equivariant task, and O(1, 3) invariant task. In O(5)-invariant regression task, we have input $X = \{x_i\}_{i=1}^2$ of type $2T_1$ and output $f(x_1, x_2) = \sin(\|x_1\|) - \|x_2\|^3/2 + \frac{x_1^T x_2}{\|x_1\|\|x_2\|}$ of type $T_0$. Then, for O(3)-equivariant task we have input $X = \{(m_i, x_i)\}_{i=1}^5$ of type $5T_0 + 5T_1$ corresponding to 5 masses

Table 2: GRpesFNOs are much cheaper than G-FNOs while giving competitive performance. Table shows mean (std) of relative mean square errors in percentage over 3 seeds. Our models use regular $T_1$ representations with the architecture kept near-identical to the FNO model used. GRepsFNO-2 uses additional early fusion layers from §. D.3 in addition to the late fusion layer present in GRepsFNO-1. Our simple architecture clearly outperforms the non-equivariant FNO architecture and performs competitively with the more sophisticated G-FNO architecture that uses group convolutions.

| Dataset \ Model | FNO | GRepsFNO-1 (ours) | GRepsFNO-2 (ours) | G-FNO |
|---|---|---|---|---|
| Navier-Stokes | 8.41 (0.41) | **5.31 (0.2)** | 5.79 (0.4) | 4.78 (0.4) |
| Navier-Stokes-Symmetric | 4.21 (0.12) | 2.92 (0.1) | **2.82 (0.1)** | 2.24 (0.1) |

and their positions. The output is the inertia matrix $\mathcal{I} = \sum_i m_i(x_i^T x_i I - x_i x_i^T)$ of type $T_2$. Finally, for the O(1, 3)-equivariant task, we consider the electron-muon scattering ($e^- + \mu^- \to e^- + \mu^-$) task used in Finzi et al. (2021), originally from Bogatskiy et al. (2020). Here, the input is of type $4T_{(1,0)}$ corresponding to the four momenta of input and output electron and muon, and the output is the matrix element (c.f. Finzi et al. (2021)), which is of type $T_{(0,0)}$.

**Experimental setup:** We train MLPs, EMLPs, GRepsNet on the datasets discussed above with varying sizes for 100 epochs. For each task and model, we choose model sizes between small (with channel size 100) and large (with channel size 384). Similarly, we choose the learning rate from $\{10^{-3}, 3 \times 10^{-3}\}$. In general, MLPs and EMLPs provide best result with the large model size, whereas GRepsNets produce better results with small model sizes. For learning rate, we find $3 \times 10^{-3}$ to be best in most cases. The exact hyperparameters used in our experiments are given in §. E.1.

**Observations and results:** From Fig. 2, we find that across all the tasks, GRepsNets perform competitively to EMLPs and significantly outperform non-equivariant MLPs. Moreover, Tab. 5 and Fig. 5 show that GRepsNet are computationally much more efficient than EMLPs, while being only slightly more expensive than naive MLPs. This shows that GRepsNet can provide competitive performance to EMLPs on equivariant tasks. Moreover, the lightweight design of GRepsNets motivates its use in larger datasets.

## 5.2 IMAGE CLASSIFICATION WITH MLP-MIXERS

**Datasets and Experimental Setup:** We work with rot90-CIFAR10 (CIFAR10 Krizhevsky et al. with random $90^0$ degree rotations), Galaxy10 Leung & Bovy, and EuroSAT Helber et al. (2018; 2019) image datasets. We compare the non-equivariant MLP-Mixers with two of our rot90-equivariant MLP-Mixers with $T_1$ representations: GRepsMLP-Mixer-1 and GRepsMLP-Mixer-2. GRepsMLP-Mixer-2 simply adds non-parametric early fusion operations in the group dimension to the GRepsMLP-Mixer-1 architecture. Each model has 8 mlp-mixer layers with patch size 16 for the Galaxy10 dataset and a patch size of 4 otherwise. Each model is trained for 100 epochs with learning rate $10^{-3}$ with 5 warmup epochs. Additional details of the architectures and hyperparameters are provided in §. E.2.

**Results and Observations:** Tab. 1 shows that GRepsMLP-Mixer models clearly outperform non-equivariant MLP-Mixer across all datasets. This indicates that even with minimal change to the original architecture, we are able to extract the benefits of group equivariance. Note that the early fusion layer in GRepsMLP-Mixer-2 helps outperform GRepsMLP-Mixer-1 in two datasets and is competitive with GRepsMLP-Mixer-1 in general. Hence, fusing the features early on helps in general, as is known in the literature (Joze et al., 2020).

## 5.3 SOLVING PDES WITH FNOS

**Datasets and Experimental Setup:** We consider two versions of the incompressible Navier-Stokes equation from Helwig et al. (2023); Li et al. (2021). The first version is a Navier-Stokes equation without any symmetry (NS dataset) in the data, and a second version that does have 90° rotation symmetry (NS-SYM dataset). The general Navier-Stokes equation considered is written as,

$$\partial_t w(x,t) + u(x,t) \cdot \nabla w(x,t) = \nu \Delta w(x,t) + f(x), \qquad (1)$$
$$\nabla \cdot u(x,t) = 0 \quad \text{and} \quad w(x,0) = w_0(x),$$

Table 3: We find that GRepsGNN provides comparable test loss and forward time compared to EGNN Satorras et al. (2021). Note that GRepsGNN is constructed by replacing the representation in the GNN architecture from Gilmer et al. (2017) with $T_1$ representations along with tensor fusion from §. 4.1, whereas EGNN are specialized GNNs designed for E($n$)-equivariant tasks.

| Model | Test Loss | Forward Time |
|---|---|---|
| EGNN | 0.0069 | 0.001762 |
| GRepsGNN (ours) | 0.0049 | 0.002018 |

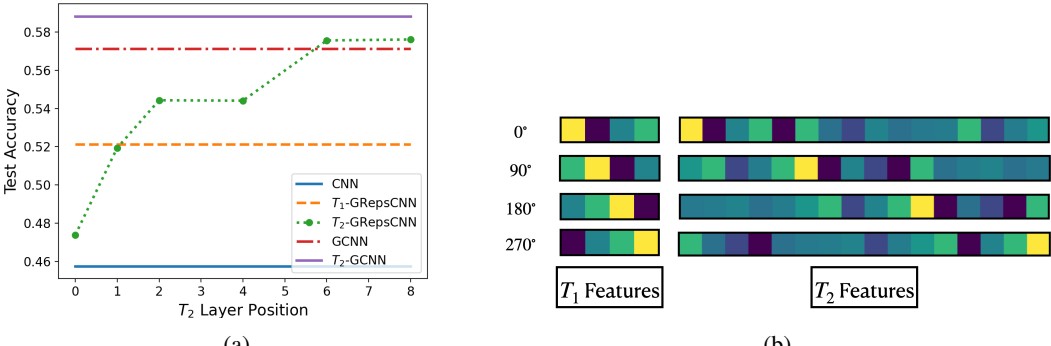

(a)                                                             (b)

Figure 3: In (a), we analyze the performance of a rot90-equivariant CNN with 3 convolutional layers and 5 fully connected layers on rot90-CIFAR10. Here, $T_2$ representations are introduced in layer $i \in [1, \ldots, 8]$. We find that using $T_2$ representations in the final layers of the CNN easily outperforms non-equivariant CNNs as well as traditional equivariant representations with $T_1$ representations. (b) shows the $T_1$ and $T_2$ features obtained from one channel of a pretrained Resnet corresponding to $T_1$-equitune and $T_2$-equitune, respectively.

where $w(x, t) \in \mathbb{R}$ denotes the vorticity at point $(x, t)$, $w_0(x)$ is the initial velocity, $u(x, t) \in \mathbb{R}^2$ is the velocity at $(x, t)$, and $\nu = 10^{-4}$ is the viscosity coefficient. $f$ denotes an external force affecting the dynamics of the fluid. The task here is to predict the vorticity at all points on the domain $x \in [0, 1]^2$ for some $t$, given the previous values of vorticity at all point on the domain for previous $T$ steps. As stated by Helwig et al. (2023), when $f$ is invariant with respect to 90° rotations, then the solution is equivariant, otherwise not. We use the same forces $f$ as Helwig et al. (2023). For non-invariant force, we use $f(x_1, x_2) = 0.1(\sin(2\pi(x_1 + x_2)) + \cos(2\pi(x_1 + x_2)))$ and as invariant force, we use $f_{inv} = 0.1(\cos(4\pi x_1) + \cos(4\pi x_2))$. We use $T = 20$ previous steps as inputs for the NS dataset and $T = 10$ for NS-SYM and predict for $t = T + 1$, same as in Helwig et al. (2023). We train our models with batch size 20 and learning rate $10^{-3}$ for 100 epochs.

**Results and Observations:** In Tab. 2, we find that both GRepsFNO-1 and GRepsFNO-2 (with early fusion) clearly outperforms traditional FNOs on both datasets NS ans NS-SYM. Note that the NS dataset do not have rot90 symmetries and yet GRepsFNOs outperforms FNOs showing that using equivariant representations may be more expressive for tasks without any obvious symmetries as was also noted in several works such as Cohen & Welling (2016a); Helwig et al. (2023). Moreover, we find that the GRepsFNO models perform competitively with the more sophisticated, recently proposed, G-FNOs. Thus, we gain benefits of equivariance with by using equivariant representations and making minimal changes to the architecture.

## 5.4 Modelling a Dynamic N-Body System with GNNs

**Datasets and Experimental Setup:** We use the N-body dynamics dataset from Satorras et al. (2021), where the task is to predict the positions of $N = 5$ charged particles after $T = 1000$ steps given their initial positions $\in \mathbb{R}^{3 \times 5}$, velocities $\in \mathbb{R}^{3 \times 5}$, and charges $\in \{-1, 1\}^5$. We closely followed Satorras et al. (2021) to generate the dataset: we used 3000 trajectories for train, 2000 trajectories

Table 4: $T_2$-equituning clearly outperforms $T_1$-equituning Basu et al. (2023b). Table shows mean (std) test accuracies for equituning (Basu et al., 2023b) using a pretrained Resnet with Rot90-CIFAR10 and Galaxy10. We find that our extension of equituning using $T_2$ representations outperforms the traditional version that only uses $T_1$ representations.

| Dataset \ Model | Finetuning | $T_1$-Equituning | $T_2$-Equituning (ours) |
|---|---|---|---|
| Rot90-CIFAR10 | 82.7 (0.5) | 88.1 (0.3) | **89.6 (0.3)** |
| Galaxy10 | 76.9 (3.2) | 79.3 (1.6) | **80.7 (4.0)** |

for validation, and 2000 for test. Both EGNN (Satorras et al., 2021) and GRepsGNN models have 4 layers, and were trained for 10000 epochs, same as in Satorras et al. (2021). Recall that GRepsGNN was designed by replacing the scalar representations in MPNN with $T_0 + T_1$ representations and their appropriate mixing.

**Results and Observations:** From Tab. 3, we note that GRepsGNN perform competitively to EGNN on the N-body problem even though EGNN is a specialized architecture for the task. Moreover, it has a comparable computational complexity to EGNN, and hence, it is computationally much more efficient than many specialized group equivariant architectures that uses spherical harmonics for $E(n)$-equivariance as noted from Tab. 6.

### 5.5 SECOND-ORDER IMAGE CLASSIFICATION

**Datasets and Experimental Setup:** We perform four sets of experiments to understand the impact of $T_2$ representations in the design of equivariant image classifiers. To that end, we first design a rot90-equivariant CNN with 3 convolutional layers followed by 5 fully connected layers and train it from scratch. We use $T_1$ representations for the first $i$ layers and use $T_2$ representations for the rest. The goal is to understand if $T_2$ representations are more useful to the network than $T_1$. This is reminiscent of the use of bilinear layers by Lin et al. (2015), where outerproduct of the features in the spatial dimension helps fine-grained classification. Here, the $T_2$ features plays a similar role, but in the group dimension and has been ignored in the literature of equivariant image classification to the best of our knowledge.

Based on the observations made, we take the equituning algorithm of Basu et al. (2023b) that uses $T_1$ representations and extend it to use $T_2$ representations in the final layers. We use pretrained Resnet18 as our non-equivariant model and perform non-equivariant finetuning and equivariant finetuning with $T_1$ and $T_2$ representations. We perform experiments on rot90-CIFAR10 and Galaxy10 datasets. Additional experimental details are provided in §. E.

**Results and Observations for Second-Order Equituning:** From Fig. 3a, we observe that using $T_2$ representations in the later layers of the same network significantly outperforms both non-equivariant and well as equivariant $T_1$-based CNNs. Finally, from Tab. 4, we find that on both rot90-CIFAR10 and Galaxy10, $T_2$-equitune easily outperforms equitune, confirming that $T_2$ features lead to powerful equivariant networks.

## 6 CONCLUSION

We present GRepsNet, a lightweight architecture designed to provide equivariance to arbitrary matrix groups like EMLPs. We find that GRepsNet gives competitive performance to EMLP on various invariant and equivariant regression tasks taken from Finzi et al. (2021), while being much more computationally affordable. Further illustrating the simplicity and generality of our design, we show that using simple first-order tensor representations in GRepsNet achieves competitive performance to specially designed equivariant networks for several different domains. We considered diverse domains such as image classification, PDE solving and $N$-body dynamics prediction using mlp-mixers, FNOs, and MPNNs, respectively, as the base model to design GRepsNet. Finally, going beyond $T_1$ representations, we show how second order tensors can be useful in image classification, which to the best of our knowledge is the first use of higher order equivariant tensors for image classification. We show that $T_2$ representation when used in the features of a CNN, provides better classification accuracies outperforming equitune.

**Reproducibility statement**   Our model design is described precisely in §. 4 and proof of its equivariance is given in §. C. Experimental settings for all our experiments and additional details on model design are detailed in §. 5, §. E, and §. D. We plan to make the code public after paper acceptance.

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

## A  ADDITIONAL DEFINITIONS

A **group** is a set $G$ along with a binary operator '$\cdot$', such that the axioms of a group are satisfied: a) closure: $g_1 \cdot g_2 \in G$ for all $g_1, g_2 \in G$, b) associativity: $(g_1 \cdot g_2) \cdot g_3 = g_1 \cdot (g_2 \cdot g_3)$ for all $g_1, g_2, g_3 \in G$, c) identity: there exists $e \in G$ such that $e \cdot g = g \cdot e = g$ for any $g \in G$, d) inverse: for every $g \in G$ there exists $g^{-1} \in G$ such that $g \cdot g^{-1} = g^{-1} \cdot g = e$.

For a given set $\mathcal{X}$, a **group action** of a group $G$ on $\mathcal{X}$ is defined via a map $\alpha : G \times \mathcal{X} \mapsto \mathcal{X}$ such that $\alpha(e, x) = x$ for all $x \in \mathcal{X}$, and $\alpha(g_1, \alpha(g_2, x)) = \alpha(g_1 \cdot g_2, x)$ for all $g_1, g_2 \in G$ and $x \in \mathcal{X}$, where $e$ is the identity element of $G$. When clear from context, we write $\alpha(g, x)$ simply as $gx$

Given a function $f : \mathcal{X} \mapsto \mathcal{Y}$, we call the function $f$ to be $G$-**equivariant** if $f(gx) = gf(x)$ for all $g \in G$ and $x \in \mathcal{X}$.

## B  ADDITIONAL DETAILS ON RELATED WORKS

### B.1  EMLPS AND UNIVERSAL SCALARS

**EMLPs**   Given the input and output types for some matrix group, the corresponding tensor representations can be derived from the given base group representation $\rho$. Using these tensor representations, one can solve for the space of linear equivariant functions directly from the obtained equivariant constraints corresponding to the tensor representations. Finzi et al. (2021) propose an elegant solution to solve these constraints by computing the basis of the linear equivariant space and construct an equivariant MLP (EMLP) from the computed basis. Our work is closest to this work as we use the same data representations as Finzi et al. (2021), but we propose a much simpler architecture for equivariance to arbitrary matrix groups. Because of the simplicity of our approach, we are able to use it for several larger datasets, which is in contrast to Finzi et al. (2021), where the experiments are mostly restricted to synthetic experiments. Moreover, using these bases are in general known to be computationally expensive (Fuchs et al., 2020).

**Universal scalars**   Villar et al. (2021) propose a method to circumvent the need to explicitly use these equivariant bases. The First Fundamental Theorem of Invariant Theory for the Euclidean group $O(d)$ states that "a function of vector inputs returns an invariant scalar if and only if it can be written as a function only of the invariant scalar products of the input vectors" (Weyl, 1946). Taking inspiration from this theorem and a related theorem for equivariant vector functions, Villar et al. (2021) characterize the equivariant functions for various Euclidean and Non-Euclidean groups. They further motivate the construction of neural networks taking the invariant scalar products of given tensor data as inputs. However, the number of invariant scalars for $N$ tensors in a data point grows as $N^2$, hence, making it an impractical method for most real life machine learning datasets. Hence, their experiments are also mostly restricted to synthetic datasets like in EMLP.

Moreover, they show (Villar et al., 2021, §. 5) that even though the number of resulting scalars grow proportional to $N^2$, when the data is of dimension $d$, approximately $N \times (d+1)$ number of these scalars is sufficient to construct the invariant function. But, it might not be trivial to find this subset of scalar for real life datasets such as images. Hence, we propose to use deeper networks with equivariant features that directly take the $N$ tensors as input, instead of $N^2$ scalar inputs, which also circumvent the need to use equivariant bases. Additional related works and comparisons are in §. 2.

## C   PROOF OF EQUIVARIANCE

Here we provide the proof of equivariance of a GRepsNet layer to matrix groups. Further, since stacking equivariant layers preserve the equivariance of the resulting model, the equivariance of the GRepsNet model follows directly. The argument is similar to the proof of equivariance of vector neurons to the $SO(3)$ group.

First consider regular representation. Note from §. D.2 and §. D.3 that the group dimension is treated like a batch dimension in regular representations for discrete groups. Thus, any permutation in the input naturally appears in the output, hence, producing equivariant output.

Now consider non-regular representations considered in the paper, e.g., matrix-based representations such as in §. 5.1. Assuming that the input to a GRepsNet layer consists of tensors of types $T_0, T_1, \ldots, T_n$, we first note that the output of the $T_0$-layer in Fig. 1a is invariant, following which we find that the $T_i$-layer outputs equivariant $T_i$ tensors.

The output of the $T_0$-layer is clearly invariant since all the inputs to the network are of type $T_0$, which are already invariant.

Now, we focus on a $T_i$-layer. Recall from §. 4.1 that a $T_i$ layer consists only of linear networks without any bias terms or pointwise non-linearities. Suppose the linear network is given by a stack of linear matrices. We show that any such linear combination performed by a matrix preserves equivariance, hence, stacking these matrices would still preserve equivariance of the output. Let the input tensor of type $T_i$ be $X \in \mathbb{R}^{c \times k}$, i.e., we have $c$ tensors of type $T_i$ and size of the representation of each tensor equals to $k$. Consider a matrix $W \in \mathbb{R}^{c' \times c}$, which multiplied with $X$ gives $Y = W \times X \in \mathbb{R}^{c' \times k}$, where $Y$ is a linear combination of the $c$ input tensors each of type $T_i$. Let the group transformation on the tensor $T_i$ be given by $G \in \mathbb{R}^{k \times k}$. Then the group transformed input is given by $X' = X \times G \in \mathbb{R}^{c \times k}$. The output of $X'$ through the $T_i$-layer is given by $Y' = W \times X \times G \in \mathbb{R}^{c' \times k} = (W \times X) \times G = Y \times G$, where the second last equality follows from the associativity property of matrix multiplication. Thus, each $T_i$-layer is equivariant.

## D   ADDITIONAL NETWORK DESIGN DETAILS

The practical design insights gained from Sec. 4.1 is described below:

1. **Regular representation**: For applications such as image classification or FNOs that usually use regular representation for equivariance Cohen & Welling (2016a); Helwig et al. (2023), we first convert the inputs to regular representations. Data represented using regular representations can be written to have a group dimension along with spatial and channel dimensions. E.g., C4 (group of 90°-rotations) $T_1$ image can be represented as a tensor of dimensions $(4, C, H, W)$, where 4 represents the size of the group dimension, and (C, H, W) represents the channel and spatial dimensions. Similarly, the C4 $T_2$ image can be represented as a tensor of dimensions $(16, C, H, W)$, where 16 represents the size of the group dimension, and (C, H, W) represents the channel and spatial dimensions. The implementation in this case simply involves treating the group dimension in the same way as the batch dimension and passing the data through arbitrary models such as CNN or MLP-Mixers as done in equituning Basu et al. (2023b) for $T_1$ representation. The use of regular representations is applicable in the cases of a) image classification with MLP-mixers, b) PDE solving with FNOs, c) equivariant image classification using CNNs.

2. **Not regular representation**: For tensor representations that are not regular representations, such as a direct sum of irreducible representations, we segregate the tensors based on their orders, then use non-linear layers on $T_0$ tensors and linear layers for higher order tensors. Moreover, the tensors are converted to different orders as described in Sec. 4.1 at each layer. This method is used for: a) synthetic experiments for comparison with EMLPs and b) predicting N-body dynamics with GNNs.

Now we provide additional details of the designed equivariant networks.

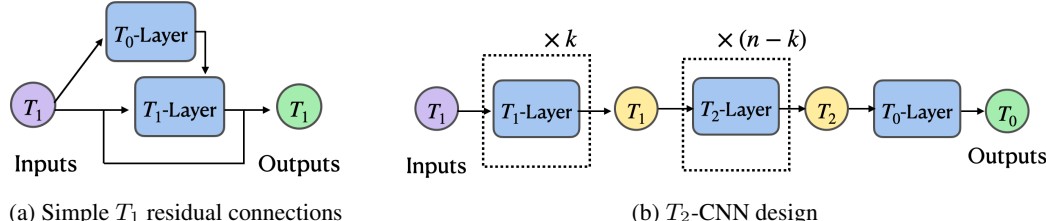

(a) Simple $T_1$ residual connections          (b) $T_2$-CNN design

Figure 4: (a) shows a simple way to add residual connections in GRepsNet. (b) shows the architecture used for $T_2$ CNNs and equituning, where the first $k$ layers are made of $T_1$-layers to extract features, then the extracted features are converted in $T_2$ tensors, which are then processed by $T_2$-layers. Finally $T_0$ tensors, i.e., scalars are obtained as the final output.

### D.1 ADDITIONAL DESIGN DETAILS FOR SYNTHETIC EXPERIMENTS RELATED TO EMLP

As mentioned in §. 4.3, we consider the three synthetic experiments from Finzi et al. (2021): $O(5)$-invariant regression task, $O(3)$-equivariant regression task, and $O(1,3)$-invariant task. Detailed description of the network designs made for these experiments are described below.

$O(5)$**-invariant model** The input consists of two tensors of $T_1$ type that is passed through the first layer consisting of $T_0$-layers and $T_1$-layers similar to vector neurons shown in Fig. 1b, but our design differs from vector neurons in that we use simple Euclidean norm to compute the $T_0$ converted tensors instead of dot product used by vector neurons. All $T_i$ layers are made of MLPs. The number of output tensors is equal to the channel size, and the channel sizes used for our experiments in discussed in §. 5.1. This is followed by three similar layers consisting of $T_0$-layers and $T_1$-layers, all of which takes as input $T_1$ tensors, and output tensors of the same type. Additionally, these layers use residual connections as shown in Fig. 4a. Finally, the $T_1$ tensors are converted to $T_0$ tensors by taking their norms, which is passed through a final $T_0$-layer that gives the output.

More precisely, in our experiments, we consider a model with 5 learnable linear layers with no bias terms, where the dimensions of the layers are $(2 \times 100, 100 \times 100, 100 \times 100, 100 \times 100, 100 \times 1)$. The input of type $2T_1$ is of dimension $(2 \times 5)$. The input is first passed through the first layer of dimension $2 \times 100$ to obtain a hidden layer output of type $100T_1$. Then, this output is also converted to type $100T_0$ by simply taking the norm. Thus, we have a tensor of type $100T_0 + 100T_1$. Finally, we convert this tensor of type $100(T_0 + T_1)$ to $100T_1$ by simply multiplying the $100T_0$ scalars with the $100T_1$ vectors. This gives a tensor of type $100T_1$, which is the input for the next layer. We repeat the same process of converting to $T_0$ and back to $T1$ for the next two layers. For the final two layers, we convert all the tensors to scalars of type $100T_0$ and process through the last two layers and use ReLU activation function in between.

$O(3)$**-equivariant model** The input consists of 5 tensors each of type $T_0 + T_1$. The first layer of our model converts them into tensors of type $T_0 + T_1 + T_2$.

A detailed description of the first layer follows. Let the input and output of the first layer be $X_{T_0}, X_{T_1}$ and $H_{T_0}, H_{T_1}, H_{T_2}$, respectively. Here, $X_{T_i}$ denotes tensors of type $T_i$ and similarly for $H_{T_i}$.

To compute $H_{T_0}$, we first convert $X_{T_1}$ to type $T_0$ by taking its norm and concatenate it to $X_{T_0}$. Let us assign this concatenated value to $H_{T_0}$. Then, the final value of $H_{T_0}$ is obtained by passing $H_{T_0}$ through two linear layers with a ReLU activation in between.

To compute $H_{T_1}$, we simply perform $W_2(H_{T_0} * W_1(X_{T_1})/\|W_1(X_{T_1})\|)$, where $W_1, W_2$ are single linear layers with no bias terms.

To compute $H_{T_2}$, we first convert $X_{T_0}$ to type $T_2$ by multiplying it with an identity matrix of dimension of $X_{T_2}$. Let us call this $H_{T_{2_0}}$. Then, we convert $X_{T_1}$ to type $T_2$ by taking the outer product with itself. Let us call this $H_{T_{2_1}}$. We concatenate $H_{T_{2_0}}$, $H_{T_{2_1}}$, and $X_{T_2}$, and call this $H_{T_2}$. Then, we update $H_{T_2}$ as follows. We simply perform $W_2(H_{T_0} * W_1(H_{T_2})/\|W_1(H_{T_2})\|)$, where $W_1, W_2$ are single-layered linear layers with no bias terms.

The number of tensors obtained is equal to the channel size used for the experiments discussed in §. 5.1. It is followed by two layers of input and output types $T_0 + T_1 + T_2$.

A detailed description of the next layers follows. Let the input and output of the first layer be $X_{T_0}, X_{T_1}, X_{T_2}$ and $H_{T_0}, H_{T_1}, H_{T_2}$, respectively. Here, $X_{T_i}$ denotes tensors of type $T_i$ and similarly for $H_{T_i}$.

To compute $H_{T_0}$, we first convert $X_{T_1}$ and $X_{T_2}$ to type $T_0$ by taking its norm and concatenate it to $X_{T_0}$. Let us assign this concatenated value to $H_{T_0}$. Then, the final value of $H_{T_0}$ is obtained by passing $H_{T_0}$ through two linear layers with a ReLU activation in between.

The rest of the computations for obtaining $H_{T_1}$ and $H_{T_2}$ are identical to the first layer, which is described below for completeness. To compute $H_{T_1}$, we simply perform $W_2(H_{T_0} * W_1(X_{T_1})/\|W_1(X_{T_1})\|)$, where $W_1, W_2$ are single linear layers with no bias terms.

To compute $H_{T_2}$, we first convert $X_{T_0}$ to type $T_2$ by multiplying it with an identity matrix of dimension of $X_{T_2}$. Let us call this $H_{T_{2_0}}$. Then, we convert $X_{T_1}$ to type $T_2$ by taking the outer product with itself. Let us call this $H_{T_{2_1}}$. We concatenate $H_{T_{2_0}}, H_{T_{2_1}}$, and $X_{T_2}$, and call this $H_{T_2}$. Then, we update $H_{T_2}$ as follows. We simply perform $W_2(H_{T_0} * W_1(H_{T_2})/\|W_1(H_{T_2})\|)$, where $W_1, W_2$ are single-layered linear layers with no bias terms.

These layers also use residual connections similar to the ones shown in Fig. 4a. Then, the $T_2$ tensors of the obtained output is passed through another $T_2$ layer, which gives the final output.

$O(1, 3)$**-invariant model** This design is identical to the design of the $O(5)$-invariant network above except for a few changes: a) the invariant tensors is obtained using Minkowski norm instead of the Euclidean norm, b) the number of channels are decided by the number of channels chosen for this specific experiment in §. 5.1.

### D.2 Additional Design Details for Image Classification using MLP-Mixer Models

Our MLP-mixers are taken from [1] contain eight layers, each containing two smaller layers: a spatial MLP-mixer layer with hidden dimension 64 and a channel MLP-mixer layer with hidden dimension 512. We provide two designs of GRepsMLP-mixers: GRepsMLP-mixer-1 and GRepsMLP-mixer-2, where GRepsMLP-mixer-1 always treats the group dimension like a batch dimension, whereas GRepsMLP-mixer-2 additionally uses a non-parametric fusion amongst the features in the group dimension. Here, we simply use a layernorm along the group dimension without any learnable parameters as our fusion layer.

We now describe the contruction of $T_1$ tensors from input images.

**Representations for image classification experiments using MLP-mixers** Given an image $x \in \mathbb{R}^{2d \times 2d}$, we can write it as $x = \begin{bmatrix} x_1 & x_2 \\ x_4 & x_3 \end{bmatrix}$, where $x_i \in \mathbb{R}^{d \times d}$ for $i \in \{1, \ldots, 4\}$. Let $G = \{e, g, g^2, g^3\}$ represent the group of $90°$ rotations. Define the group action of $G$ on $x$ naturally, i.e. $gx = \begin{bmatrix} gx_4 & gx_1 \\ gx_2 & gx_3 \end{bmatrix}$.

We use the following $G$ representation of $x$ as $(x)_G = \begin{bmatrix} x_1 & g^{-1}x_2 \\ g^{-3}x_4 & g^{-2}x_3 \end{bmatrix}$. Each of the four entries in the matrix $(x)_G$ are treated as separate channels with no data flowing between except when intra-mixers are used. Further all the channels share the same parameters, say $\mathrm{M}$. Then the output of $\mathrm{M}$ would be $\mathrm{M}((x)_G) = \begin{bmatrix} \mathrm{M}(x_1) & \mathrm{M}(g^{-1}x_2) \\ \mathrm{M}(g^{-3}x_4) & \mathrm{M}(g^{-2}x_3) \end{bmatrix}$. We now verify the equivariance obtained using this representation. First, note $(gx)_G = \begin{bmatrix} g^{-3}x_4 & x_1 \\ g^{-2}x_3 & g^{-1}x_2 \end{bmatrix}$. Then, the output is $\mathrm{M}((gx)_G) = \begin{bmatrix} \mathrm{M}(g^{-3}x_4) & \mathrm{M}(x_1) \\ \mathrm{M}(g^{-2}x_3) & \mathrm{M}(g^{-1}x_2) \end{bmatrix}$. Clearly, $\mathrm{M}((gx)_G)$ is a permutation of $\mathrm{M}((x)_G)$. To obtain invariance, we simply average the four channels. This method is computationally more efficient than using four transformed images as input as done in Basu et al. (2023b). We find this simple and efficient representation is still able to gain the benefits of group equivariance.

---

[1]https://github.com/omihub777/MLP-Mixer-CIFAR

### D.3 Additional Design Details for PDE solving using FNOs

Here we describe the construction of the $T_1$ representation used. We directly use the $T_1$ representation from equitune (Basu et al., 2023b). We do not use the same $T_1$ representations used for MLP-mixers since the group dimension is treated like the batch dimension, which effectively reduces the image dimension. This affects the available frequencies in the Fourier domains of the inputs to the FNOs. Thus, we choose the naive $T_1$ representations of equitune here that naturally preserves all the frequencies since the effective image dimensions remain the same.

We provide two designs of GRepsFNO, similar to GRepsMLP-mixers: GRepsFNO-1 and GRepsFNO-2, where GRepsFNO-1 always treats the group dimension like a batch dimension, whereas GRepsFNO-2 additionally uses a non-parametric fusion amongst the features in the group dimension. Here, we simply divide the features in the group dimension by the standard deviation across that dimension. We avoid using layernorm like in GRepsMLP-mixers here since it requires implementation of layernorm in the complex Fourier domain. Instead, the choice of dividing by the standard deviation is a much simpler alternative serving the same purpose.

**Representations for solving PDEs using FNOs** For FNOs, we use the traditional group representations as used in Basu et al. (2023b), i.e. $(x)_G = \begin{bmatrix} x & gx \\ g^3x & g^2x \end{bmatrix}$. The reason for our choice here are as follows: a) for FNOs, we preserve all the frequency modes by using transformed inputs, b) for CNNs, we consider the setup of equituning and extend it to second-order representations, hence, using the representation here gives a more direct comparison.

## E Additional Experimental Details

### E.1 Comparison with EMLPs

Here we provide the learning rate and model sizes used for the experiments on comparison with EMLPs in §. 5.1.

In the O(5)-invariant regression task, for MLPs and EMLPs, we use a learning rate of $3 \times 10^{-3}$ and channel size $384$. Whereas for GRepsNets, we use a learning rate of $10^{-3}$ and channel size $100$.

For the O(3)-equivariant task, we use learning rate $10^{-3}$ and channel size $384$ for all the models.

For the O(1, 3)-invariant regression task, we use a learning rate of $3 \times 10^{-3}$ for all the models. Further, we use a channel size of $384$ for MLPs and EMLPs, whereas for GRepsNets, a channel size of $100$ was chosen as it gives better result.

### E.2 Image Classification with MLP-Mixers

In each of the models, every layer further consists of two smaller layers: a) one that applies a layernorm followed by two-layered MLP on the channel dimension, and b) another that applies layernorm followed by two-layered MLP on the spatial dimension, as done in traditional MLP-Mixers (Tolstikhin et al., 2021). For GRepsMLP-Mixer-1, we simply replace the traditional scalar representations of MLP-Mixers by $T_1$ representations described in §. D.2. For GRepsMLP-Mixer-2, we additionally add early fusion layer with no additional parameters. The early fusion layer is constructed as follows: the four different components of the $T_1$ representation for the rot90 group is fused using a simple layernorm applied along this *group dimension* of the $T_1$ representation of size four.

For training our non-equivariant and equivariant MLP-Mixer models, we closely follow the training setup of [2] and train each of the models for 100 epochs. In particular, we train each model with batch size 128 for 100 epochs with learning rate $10^{-3}$ and use 5 warmup epochs with minimum learning rate $10^{-6}$, use cosine scheduler (Loshchilov & Hutter, 2017) and Adam optimizer (Kingma & Ba, 2015) with $(\beta_1, \beta_2) = (0.9, 0.99)$, weight decay $5 \times 10^{-5}$.

---

[2]https://github.com/omihub777/MLP-Mixer-CIFAR

### E.3 SECOND-ORDER IMAGE CLASSIFICATION

**Training CNNs from scratch:** The CNN used for training from scratch consists of 3 convolutional layers each with kernel size 5, and output channel sizes 6, 16, and 120, respectively. Following the convolutional layers are 5 fully connected layers, each consisting of features of dimension 120. For training from scratch, we train each model for 10 epochs, using stochastic gradient descent with learning rate $10^{-3}$, momentum 0.9. Further, we also use a stepLR learning rate scheduler with $\gamma$ 0.1, step size 7, which reduces the learning rate by a factor of $\gamma$ after every step size number of epochs. The $T_2$ layers are computed by simply taking an outer product of the $T_1$ features at the desired layer where $T_2$ representation is introduced, following which, we simply use the same architecture as for $T_1$ representation. It is easy to verify the equivariance is maintained for both $T_1$ and $T_2$ for regular representations.

**Second-Order Finetuning:** For finetuning the pretrained Resnet18, we use 5 epochs, using stochastic gradient descent with learning rate $10^{-3}$, momentum 0.9. For equivariant finetuning with $T_2$ representations, we first extract $T_1$ featured from the pretrained model same as done for equituning (Basu et al., 2023b), following which we convert it to $T_2$ representations using a simple outer product. Once the desired features are obtained, we pass it through two fully connected layers with a ReLU activation function in between to obtain the final classification output.

**Comparison to GCNNs and E(2)-CNNs:** For each of the models, we use 3 convolutional layers (either naive convolutions or group convolutions) followed by 5 fully connected layers. For all variants of GRepsCNNs, the convolutional layers consist of kernel sizes 5, channel sizes 6, 16, and 120. The linear layers are all of dimension $120 \times 120$ with ReLU activations in between and residual connections. All GRepsCNNs consist of 153k parameters. For the GCNNs, we reduce the channel sizes to 3, 10, 100, and the linear layers to be of dimension $100 \times 100$ to adjust the number of parameters, which amounts to 175k. For E(2)-CNNs, we keep the channel sizes the same as GRepsCNNs since we found the performance drop when reducing channel sizes. Thus, the resulting models for C8-E(2)CNNs and C16-E(2)CNNs have 280k and 458k params, respectively. The hyperparameters are kept the same as for training the CNNs from scratch.

We also provide comparisons with baseline architectures such as GCNNs Cohen & Welling (2016a) and E(2)-CNNs Weiler & Cesa (2019). First, for comparison with GCNNs, we take the same CNN architecture from above with 3 convolutional layers followed by 5 fully connected layers. We design two GRepsCNN architectures: a) $T_1$-GRepsCNN, where each layer has a $T_1$ representation, and b) $T_2$-GRepsCNN, where all the layers except the last layer use $T_1$ representation and the last layer uses $T_2$ representation. Similarly, we design two GCNN architectures, $T_1$-GCNN and $T_2$-GCNN. Both architectures consist of 3 group convolutions followed by 5 layers of fully connected layers. Here the group convolutions correspond to the C4 group of 90-degree rotations. We call the original GCNN proposed by Cohen & Welling (2016a) as $T_1$-GCNN since all the features are $T1$. Additionally, we introduce a new variant of GCNN that we call $T_2$-GCNN, which uses the same GCNN architecture, except that it uses a $T_2$ feature in the final layer. These models are compared on the Ro90-CIFAR10 dataset.

For comparison with E(2)-CNNs, we perform a similar comparison as that with GCNNs. Here we work with Rot-CIFAR10 dataset, where the CIFAR10 dataset is rotated by random angles in $(-180°, 180°)$. This is because $E(2)$-CNNs are equivariant to larger groups than that of 90-degree rotations, so, we want to test the model's capabilities for these larger group symmetries. Here we build four variants of GRepsCNNs: C8-$T_1$-GRepsCNN, C8-$T_2$-GRepsCNN, C16-$T_1$-GRepsCNN, and C16-$T_2$-GRepsCNN. Here C$N$ for $N \in \{8, 16\}$ corresponds to the groups of $\frac{360}{N}$-degree rotations. All the GRepsCNN consist of 3 layers of convolutions followed by 5 fully connected layers. The layer representations for C$N$-$T_1$-GRepsCNN are all $T_1$ tensors of the C$N$ group. Whereas for C$N$-$T_1$-GRepsCNN, all layer representations except the last layer are $T_1$ representations and the last layer uses $T_2$ representations. The input representation of type $T_1$ for all the GRepsCNNs here are directly constructed from the E(2)-CNN paper Weiler & Cesa (2019) and the $T_2$ representations are obtained by simple outer product of the $T_1$ representation in the group dimension. Similar to GCNNs, we construct two types of E(2)-CNNs, of types $T_1$ and $T_2$, for each group C$N$, $N \in \{8, 16\}$. C$N$-$T_1$-E(2)-CNN is the traditional E(2)-CNN for the C$N$ group, which has $T_1$ representation at each layer. C$N$-$T_2$-E(2)-CNN has $T_1$ representations for each layer except for the last layer that uses a $T_2$ representation.

Table 5: EMLP is computationally extremely expensive compared to MLPs and GRepsNet. Train time per epoch (in seconds) for models with the same channel size of 384 for datasets of size 1000. GRepsNets provide the same equivariance as EMLPs but at a much affordable compute cost that is comparable to MLPs. Thus, EMLPs, despite their excellent performance on equivariant tasks, is not scalable to larger datasets of practical importance.

| Model \ Task | O(5)-invariant | O(3)-equivariant | SO(1, 3)-invariant |
|---|---|---|---|
| MLP | 0.0083 | 0.0087 | 0.008 |
| GRepsNet | 0.013 | 0.084 | 0.049 |
| EMLP | 3.00 | 3.19 | 2.86 |

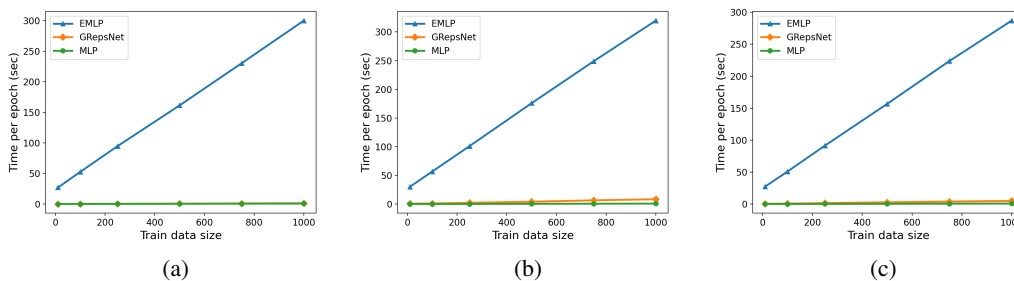

(a) (b) (c)

Figure 5: Times per epoch (in seconds) for different MLPs, GRepsNets, and EMLPs for varying dataset sizes. Note that MLPs and GRepsNets have comparable time per epoch, whereas EMLPs take huge amount of time. Hence, EMLPs, despite its excellent performance on equivariant tasks, is not scalable to larger datasets of practical importance.

## F    ADDITIONAL RESULTS

**Comparison of training time with EMLP**    Fig. 5 compares the training times takes by EMLPs, GRepsNets, and MLPs per epoch for varying dataset sizes on the three datasets considered in §. 5.1. Results show that increasing the size of the train data significantly increases the training time for EMLPs, whereas for MLPs and GRespNets, the increase in training time, although linear, is negligible. It shows that GRepsNet are more suitable for equivariant tasks for larger datasets.

**Comparison of time for forward passes in GNN models**    We present the results of forward pass times for various equivariant and non-equivariant graph neural network models in Tab. 6 taken directly from Satorras et al. (2021). It shows that networks constructed from equivariant bases such as tensor field networks (TFNs) and SE(3)-equivariant transformers can be significantly slower than non-equivariant graph neural networks.

Table 6: We know from previous results that EGNN is much faster than other equivariant networks such as SE (3) Transformers and outperforms them in test loss performance. Results taken from Satorras et al. (2021).

| Model | Test Loss | Forward Time |
|---|---|---|
| Linear | 0.0819 | 0.0001 |
| SE (3) Transformer | 0.0244 | 0.1346 |
| Tensor Field Network | 0.0155 | 0.0343 |
| Graph Neural Network | 0.0107 | 0.0032 |
| Radial Field Network | 0.0104 | 0.0039 |
| EGNN | 0.0071 | 0.0062 |

Table 7: GRpesFNOs are much cheaper than G-FNOs while giving competitive performance. Table shows mean forward time (in seconds) per epoch over 5 epochs for FNO, GRepsFNO-1, GRepsFNO-2, and G-FNO models on Navier-Stokes and Navier-Stokes-Symmetric datasets as described in Sec. 5.3.

| Dataset \ Model | FNO | GRepsFNO-1 (ours) | GRepsFNO-2 (ours) | G-FNO |
|---|---|---|---|---|
| Navier-Stokes | 49.8 | 53.9 | 70.3 | 109.9 |
| Navier-Stokes-Symmetric | 19.2 | 20.8 | 23.2 | 43.8 |

Table 9: $T_2$-GCNN outperforms the traditional GCNN. $T_2$-GRepsCNN also performs competitively, showing the importance of use of higher order tensors. Table shows mean (std) of classification accuracies on Rot90-CIFAR10 dataset for $T_1$-GRepsCNN, $T_2$-GRepsCNN, $T_1$-GCNN and $T_2$-GCNN for 10 epochs. Results are over 3 seeds.

| Dataset \ Model | CNN | $T_1$-GRepsCNN | $T_2$-GRepsCNN | $T_1$-GCNN | $T_2$-GCNN |
|---|---|---|---|---|---|
| Rot90-CIFAR10 | 46.6 (0.8) | 53.3 (1.4) | **57.9 (0.6)** | 56.7 (0.5) | **57.9 (0.7)** |

**Comparison of time for forward passed in FNO models** In Tab. 7, we provide the forward times for the various FNO models considered in Sec. 5.3 for Navier-Stokes and Navier-Stokes-Symmetric datasets. We find that G-FNO takes significantly more time than naive FNOs. Whereas GRepsFNO-1s take almost the same time as FNOs and GRepsFNO-2 takes a little more time than GRepsFNO-1. Overall, from Tab. 2 and 7, we conclude that GRepsFNO models can provide a significant advantage over FNOs with little computational overhead.

**Results and Observations for Comparison to GCNN and E(2)-CNNs:** From Tab. 9 and 10, we make two key observations: a) $T_2$-GRepsCNNs are competitive and often outperform the baselines GCNNs and E(2)-CNNs, b) $T_2$ features, when added to the baselines to obtain $T_2$-GCNNs and C$N$-$T_2$-E(2)-CNNs, they outperform their original outperform. This shows the importance of higher-order tensors in image classification. Thus, we not only provide competitive performance to baselines using our models, but also improve the results from these baselines by adding $T_2$ features in them.

## G ON THE UNIVERSALITY OF THE GREPSNET ARCHITECTURE

We provide simple constructive proofs showing the universality properties of the GRepsNet architecture.

We first show that GRepsNet can approximate arbitrary invariant scalar functions of vectors from $O(d)$ and $O(1, d)$ groups. Then, we extend the proof for vector-valued functions for the same groups.

First, recall the Fundamental Theorem of Invariant Theory for O(d) as described in Lemma .1.

**Lemma 1** (Weyl (1946)). *A function of vector inputs returns an invariant scalar if and only if it can be written as a function only of the invariant scalar products of the input vectors. That is, given input vectors $(X_1, X_2, \ldots, X_n)$, $X_i \in \mathbb{R}^d$, any invariant scalar function $h : \mathbb{R}^{d \times n} \mapsto \mathbb{R}$ can be written as*

$$h(X_1, X_2, \ldots, X_n) = f(\langle X_i, X_j \rangle_{i,j=1}^n), \tag{2}$$

*where $\langle X_i, X_j \rangle$ denotes the inner product between $X_i$ and $X_j$, and $f$ is an arbitrary function.*

As mentioned in Villar et al. (2021), a similar result holds for the $O(1, d)$ group. In Thm. 1, we show that GRepsNet can approximate arbitrary invariant scalar functions for $O(d)$ or $O(1, d)$ groups.

**Theorem 1.** *For given $T_1$ inputs $(X_1, X_2, \ldots, X_n)$ corresponding to $O(d)$ or $O(1, d)$ group, $X_i \in R^d$, any invariant scalar function $h : \mathbb{R}^{d \times n} \mapsto \mathbb{R}$, there exists a GRepsNet model that can approximate $h$.*

*Proof.* Let the tensors of type $i$ at layer $l$ be written as $H_i^l$. Given input $(X_1, X_2, \ldots, X_n) \in \mathbb{R}^d$ of type $T_1$, we construct a GRepsNet architecture that can approximate $h$ by taking help from the

Table 10: $T_2$-E(2)CNN outperforms the traditional E(2)CNN. $T_2$-GRepsCNN also perform competitively, showing the importance of use of higher order tensors. Table shows mean (std) of classification accuracies on Rot-CIFAR10 dataset (CIFAR10 with random rotations in (-180°, +180°]) for various GRepsCNNs and E(2)-CNNs with different group equivariances, and tensor orders. All models are trained for 10 epochs and results are over 3 fixed seeds.

| Model | Equivariance | Tensor Orders | Test Acc. |
|---|---|---|---|
| CNN | – | – | 43.2 (0.2) |
| C8-$T_1$-GRepsCNN | C8 | $(T_1)$ | 48.1 (1.2) |
| C8-$T_2$-GRepsCNN | C8 | $(T_1, T_2)$ | 53.1 (0.7) |
| C8-$T_1$-E(2)CNN | C8 | $(T_1)$ | 51.7 (1.8) |
| C8-$T_2$-E(2)CNN | C8 | $(T_1, T_2)$ | **54.3 (1.2)** |
| C16-$T_1$-GRepsCNN | C16 | $(T_1)$ | 48.4 (1.4) |
| C16-$T_2$-GRepsCNN | C16 | $(T_1, T_2)$ | 53.6 (0.7) |
| C16-$T_1$-E(2)CNN | C16 | $(T_1)$ | 50.6 (1.6) |
| C16-$T_2$-E(2)CNN | C16 | $(T_1, T_2)$ | **53.7 (1.2)** |

approximation properties of a multi-layered perceptron Hornik et al. (1989). Let the first layer consist only of $T_1$-layers, i.e., linear layers without any bias terms such that the obtained hidden layer $H_1^1$ is of dimension $\mathbb{R}^{d \times (n^2+n)}$ and consists of the $T_1$ tensors $X_i + X_j$ for all $i, j \in \{1, \ldots, n\}$ and $X_i$ for all $i \in \{1, \ldots, n\}$. This can be obtained by a simple linear combination. Now, construct the second layer by first taking the norm of all the $T_1$ tensors, which gives $\|X_i\| + \|X_j\| + 2 \times \langle X_i, X_j \rangle$ for all $i, j \in \{1, \ldots, n\}$ and $\|X_i\|$ for all $i \in \{1, \ldots, n\}$. Then, using a simple linear combination of the converted $T_0$ tensors give $\langle X_i, X_j \rangle$ for all $i, j \in \{1, \ldots, n\}$. Finally, passing $\langle X_i, X_j \rangle$ for all $i, j \in \{1, \ldots, n\}$ through an MLP gives $H_0^2$. Now, from the universal approximation capability of MLPs, it can approximate $f$ from equation 3. Thus, we obtain the function $h$ from Lem. 1. □

Now, recall from Villar et al. (2021), a statement similar to Lem. 1, but for vector functions.

**Lemma 2** (Villar et al. (2021)). *A function of vector inputs returns an equivariant vector if and only if it can be written as a linear combination of invariant scalar functions times the input vectors. That is, given input vectors $(X_1, X_2, \ldots, X_n)$, $X_i \in \mathbb{R}^d$, any equivariant vector function $h : \mathbb{R}^{d \times n} \mapsto \mathbb{R}^d$ can be written as*

$$h(X_1, X_2, \ldots, X_n) = \sum_{t=1}^{n} f_t(\langle X_i, X_j \rangle_{i,j=1}^n) X_t, \tag{3}$$

*where $\langle X_i, X_j \rangle$ denotes the inner product between $X_i$ and $X_j$, and $f_t$s are some arbitrary functions.*

Now, in Thm. 2 we prove the universal approximation capability of GRepsNet architecture for vector functions.

**Theorem 2.** *For given $T_1$ inputs $(X_1, X_2, \ldots, X_n)$ corresponding to $O(d)$ or $O(1, d)$ group, $X_i \in R^d$, any equivariant vector function $h : \mathbb{R}^{d \times n} \mapsto \mathbb{R}^d$, there exists a GRepsNet model that can approximate $h$.*

*Proof.* The proof closely follows the proof for Thm. 1. Let the tensors of type $i$ at layer $l$ be written as $H_i^l$. Given input $(X_1, X_2, \ldots, X_n) \in \mathbb{R}^d$ of type $T_1$, we construct a GRepsNet architecture that can approximate $h$ by taking help from the approximation properties of a multi-layered perceptron Hornik et al. (1989). Let the first layer consist only of $T_1$-layers, i.e., linear layers without any bias terms such that the obtained hidden layer $H_1^1$ is of dimension $\mathbb{R}^{d \times (n^2+n)}$ and consists of the $T_1$ tensors $X_i + X_j$ for all $i, j \in \{1, \ldots, n\}$ and $X_i$ for all $i \in \{1, \ldots, n\}$. This can be obtained by a simple linear combination. Let the second layer consist of both a $T_0$ layer and a $T_1$ layer. Let the $T_0$ layer output, $H_0^1$, be $\langle X_i, X_j \rangle$ for all $i, j \in \{1, \ldots, n\}$ and $\|X_i\|$ for $i \in \{1, \ldots, n\}$ in a similar way as done in the proof for Thm. 1. And let the $T_1$ layer output, $H_1^1$, be $X_i$ for $i \in \{1, \ldots, n\}$. Again, let the

third layer also consist of a $T_0$ layer and a $T_1$ layer. Let the $T_0$ layer consist of MLPs approximating the output $\|X_t\| \times f_t(\langle X_i, X_j \rangle_{i,j=1}^n)$ for $t \in \{1, \ldots, n\}$. Denote $\|X_t\| \times f_t(\langle X_i, X_j \rangle_{i,j=1}^n)$ as $H_0^{3,t}$. Then, let the $T_1$ layer consist of first mixing the scalars $H_0^{3,t}$ with $X_t$ as described in Sec. 4.1 as

$$H_1^{3,t} = X_t \times \frac{H_0^{3,t}}{\|X_t\|},$$

where $H_1^{3,t}$ for $t \in \{1, \ldots, n\}$ represent the output of the $T_1$ layer of the third layer. Note that from the universal approximation properties of MLPs Hornik et al. (1989), we get that $H_1^{3,t}$ approximates $X_t \times f_t(\langle X_i, X_j \rangle_{i,j=1}^n)$. Finally, the fourth layer consists of a single $T_1$ layer that sums the vectors $H_1^{3,t}$ for $t \in \{1, \ldots, n\}$, which combined with Lem. 2 concludes the proof. $\qquad \square$

Thus, we find that a simple architecture can universally approximate invariant scalar and equivariant vector functions for the $O(d)$ or $O(1, d)$ groups. This is reminiscent of the universality property of a single-layered MLP. However, in practice, deep neural networks are known to have better representational capabilities than a single-layered MLP. In a similar way, in practice, we design deep equivariant networks using the GRepsNet architecture that provides good performance on a wide range of domains.

