# OpenReview forum: "GRepsNet: A Simple Equivariant Network for Arbitrary Matrix Groups"
_ICLR.cc/2024/Conference — Submitted to ICLR 2024_

### Official Review · Reviewer_6e6R · 2023-10-22

**Soundness:** 2 fair
**Presentation:** 2 fair
**Contribution:** 2 fair
**Rating:** 3
**Confidence:** 3

**Summary:**

This paper construct equivariant neural networks for arbitrary matrix groups. The core insight is that tensors of type $i$ with $i = kj + r$ ($i > j > 0$) can be obtained by taking $T_j^{\otimes k} \otimes T_1^{\otimes r}$. In this way, higher-order tensors can be constructed from lower-order tensors and mixing is then done through linear layers. Further, scalars (type-0 tensors, invariants) can further interact nonlinearly with the higher-order tensors. This efficient architecture is tested on several experiments including synthetic experiments from Finzi et al. (2021), image classification, PDE solving, $n$-body dynamics, and equivariant fine-tuning. Results show competitive performances using a lower computational budget.

**Strengths:**

* The paper proposes a simple, effective and efficient method for equivariant neural networks on arbitrary matrix groups.
* The search for efficient and effective equivariant architectures is still ongoing, making it a timely and significant contributions.
* The method is tested on several distinct experiments.

**Weaknesses:**

See questions for further elaborations.

* Clarity and quality: there are several important details missing regarding implementations and experimental details.
* The claim is that the method is equivariant to arbitrary matrix groups, but only subgroups of the orthogonal group are considered in the experiments.
* Several relevant baselines are missing for e.g. the $n$-body experiments. For example, Brandsetter et al., 2022 and Ruhe et al., 2023 are currently state of the art on the $n$-body experiments but are left out of the picture.
* The Finzi et al. (2021) experiments are altered to a low-data setting.
* It is unclear how second-order tensor-features are used in the image classification experiments. Further, sentences like "GRepsMLP-Mixer-2 simply adds non-parametric early fusion operations in the group dimension to the GREpsMLP-mixer-1 architecture" are completely opaque to me. More care needs to be taken to explain what the authors do exactly.

Minor:
* Repeatedly, the *vector neurons* architecture is called $\mathrm{SO}(3)$-equivariant, and while the original paper frames itself as such, it is technically $O(n)$-equivariant. As such, it is less restricted than sometimes claimed.

**Questions:**

* Why did you limit yourself to orthogonal group experiments where e.g. Finzi et al. (2021) also consider other matrix groups?
* It is not quite clear how equivariance is achieved in the image domain, where usually equivariant architectures consider images as maps from positions to (RGB) values. Inspecting the appendix, I noted that the authors repeat images four times in 90-degree rotations. However, isn't this a different way of achieving equivariance compared to what the main method proposes? The main method combines tensors of different types linearly and then nonlinearly using equivariant nonlinearities, but I don't see how that is used in the image case. Similarly for the FNO comparison.
* What do the equivariant layers exactly look like. Could you write down what the conversion layers and linear layers comprise mathematically?
* How is equivariance achieved for the image experiments? Could you provide more details into your architectures, experimental setup, and so on?
* Why do $N$ tensors lead to $N^2$ invariants?
* Could you discuss the limitations of your method? Does it replace all tensor-based methods or does it have specific use-cases? Is it as expressive as previous methods? It seems that there are no bilinear operations (e.g., tensor products) in your architectures, is that a problem?
* The authors bolden their numbers even though some method might be better than theirs but uses a more sophisticated technique. E.g., for G-FNO in Table 3. Instead, I would also provide the forward/backward times and include a comparison on that level, as efficiency is a core part of the method.
* Why does the number of data-points in your synthetic experiments not extend beyond $~1000$? In Finzi et al., settings up to 30 000 datapoints are considered.

---

> ### Author Response · Authors · 2023-11-22
> **Response to reviewer 6e6R**
>
> We thank the reviewer for their detailed feedback. We answer all the concerns raised below.
>
> **Reviewer:** Why did you limit yourself to orthogonal group experiments where e.g. Finzi et al. (2021) also consider other matrix groups?
>
> **Response:** Please note that our model design is inspired from the EMLP model (Finzi et al. (2021)). Our formulation can handle general matrix groups just like EMLP. Hence, we closely followed many of the experiments of EMLPs to provide a fair comparison with EMLP.
> Please note that Finzi et al. also provides experiments only corresponding to orthogonal groups in Sec. 7 of their work. In Fig. 7 of their work, Finzi et al. only provides the construction of their convergence algorithm for non-orthogonal groups, but do not provide any experiments for performance evaluation corresponding to those constructions.
>
> **Reviewer:**  It is not quite clear ... I don't see how that is used in the image case. Similarly for the FNO comparison.
>
> **Response:** For any data domain (e.g., images, graphs, pdes), we assume that the input to the network is a tensor of certain known type for some known group. For example, 5 (T_0 + T_1) for the O(5) used in some experiments in Sec. 5.1. Similarly, for image datasets in Sec. 5.2 and FNO datasets in Sec. 5.3, we assume that the inputs are first preprocessed to obtain data of type T_1 corresponding to the C4 group of 90-degree rotations.
>
> This preprocessing can be performed in many ways, such as rotating parts of the inputs as we do for images or rotating entire images as done for FNOs as described in Sec. D.2. Note that any preprocessing that outputs T_1 representation would work for our experiments. Also note that for images, we first divide the image into 4 parts from the center and rotate each with different multiple of 90 to obtain a T1 representation of image, where this T1 representation has the same size as the original image. On the other hand, for FNO, we simply rotate the entire image to obtain the T1 representations. This is because dividing the network into smaller parts would affect the frequencies considered for Fourier transform in the FNO. In both these cases, we simply worked with T1 representation across the network.
>
> For second order finetuning in Sec. 5.5, we first use T1 representation, followed by their conversion to T2 representation. We find that T2 representation when used with features obtained from pretrained models gives better classification performance when compared simply using T1 representation. The motivation here is similar to [1] where outer product in the space dimension helps provide better features. Instead, we take outer product in the group dimension resulting in a T2 representation that is equivariant to the same group C4.
>
> [1] Lin et al. "Bilinear cnn models for fine-grained visual recognition." ICCV
>
> **Reviewer:** What do the equivariant layers exactly look like. Could you write down what the conversion layers and linear layers comprise mathematically?
>
> **Response:** We have now updated the details of our architecture in Sec. D.1 to describe how exactly we form the conversion and linear layers. In short, the key to the working of our architecture lies in the representation of the input, output and hidden layers.
> Input representation: In Sec. 5.1, 5.4 we are given the input in the form of tensor representations, whereas, in Sec. 5.2, 5.3, and 5.5 we convert the inputs (images or pdes) to appropriate tensor representations before feeding into the network.
>
> Hidden layer and output representations: In each of the applications, the networks are designed based such that the hidden representations are of appropriate tensor types. All details are now provided in Sec. D.1.
>
> **Reviewer:** How is equivariance achieved for the image experiments? Could you provide more details into your architectures, experimental setup, and so on?
>
> **Response:** Please note that for images (as well as PDEs), we convert the input to T1 tensors corresponding to the regular representation of the C4 group (please check the details in Sec. D.2, D.3). For the image experiments with MLP-MIxers in Sec. 5.2, we keep T1 tensors for all the hidden representations, except for the output that is converted to T0 tensors. Whereas for the finetuning experiments in Sec. 5.5, we start with regular T1 representations, then convert them to T2 representations to obtain better features as explained above. We don't find much advantage with the use of T2 representations with MLP-Mixers features, hence, we only use them with CNNs.
>
> **Reviewer:** Why do N  tensors lead to N^2  invariants?
>
> **Response:** Because in the First Fundamental Theorem of Invariant Theory for the Euclidean group O (d) [2], the invariants correspond to the inner products of input vectors. Hence, for N input vectors, we have N^2 inner products or invariants. We hope this clarifies the question.
>
> [2] Weyl. The classical groups: their invariants and representations, 1946.

---

> > ### Author Response · Authors · 2023-11-22
> > **(Contd.) Response to reviewer 6e6R**
> >
> > **Reviewer:** Could you discuss the limitations of your method? Does it replace all tensor-based methods or does it have specific use-cases? Is it as expressive as previous methods? It seems that there are no bilinear operations (e.g., tensor products) in your architectures, is that a problem?
> >
> > **Response:** Our method is not intended to replace all tensor-based methods. Our method is intended to provide a simple yet expressive method to design a wide range of group equivariant networks with minimal restrictions in the choice of the domain or the base model (such as CNN or MLP-Mixers).
> >
> > The ability to work with different base models while preserving equivariance and having data corresponding to any desired tensor types is crucial. This circumvents the need to design novel equivariant models for new base models arising in the ever growing research area of deep learning. This way, researchers can directly plug new base models and use any data types required for any domain.
> >
> > In general, our model is very expressive. Indeed, on the requests by reviewers rnny, YzuG, we now add a simple constructive proof in Sec. G in the appendix showing that GRepsNet has universality properties for scalar and vector functions with vector inputs for the O(d) and O(1, d) group. This shows that our model is indeed very expressive for a wide range of invariant and equivariant tasks.
> >
> > Bilinear layers could certainly help the performance of our design. However, we intended to keep the model simple and given the universality proof (now added), we believe that using a larger base model can easily compensate for any performance drop because of lack of more complex layers.
> >
> > **Reviewer:** The authors bolden their numbers even though some method might be better than theirs but uses a more sophisticated technique. E.g., for G-FNO in Table 3. Instead, I would also provide the forward/backward times and include a comparison on that level, as efficiency is a core part of the method.
> >
> > **Response:** We have now added the forward times for all the FNO models (i.e., FNO, GRepsFNO-1, GRepsFNO-2, G-FNO) for the considered Navier-Stokes and Navier-Stokes-Symmetric datasets in Tab.7 in Sec. F of the paper. Note that GRepsFNOs provide significant performance gain over FNOs with little computational overhead. Further, GRepsFNOs are very simple to design compared to the more sophisticated G-FNOs.
> >
> > **Reviewer:** Why does the number of data-points in your synthetic experiments not extend beyond 1000 In Finzi et al., settings up to 30 000 datapoints are considered.
> >
> > **Response:** Please note that for our synthetic experiments, we had to limit ourselves to 1000 data points since we found EMLPs to be computationally expensive as shown in Fig. 5 of our paper. And due to computational overhead of EMLPs, we are unable to provide experiments up to 30,000 datapoints. Also, please note from Fig. 5 from Finzi et al. that the loss values drop very little after 1000 datapoint for 2 of the 3 experiments. Moreover, in the synthetic experiments, our main goal was simply to show that one can design simple architectures such as GRepsNet that can compete with much sophisticated architectures such as EMLPs.

---

> > > ### Comment · Reviewer_6e6R · 2023-11-22
> > >
> > > Thank you for your detailed feedback. Unfortunately, I still think the paper is not yet in a state ready to be published, hence my score remains. I encourage the authors to resubmit to another upcoming high-impact venue. Note also that current version's page limit is far exceeded.

---

> > > > ### Author Response · Authors · 2023-11-22
> > > > **Response to official comment by Reviewer 6e6R**
> > > >
> > > > We thank the reviewer for their prompt response.
> > > >
> > > > We apologize for mistakenly exceeding the page limit while updating our draft to add clarifications and experiments requested by all the reviewers.
> > > >
> > > > We have now updated the draft to add the requested clarifications/experiments in the appendix to ensure the page limit is not exceeded.
> > > >
> > > > We believe we have addressed all the major concerns raised by the reviewer. We would highly appreciate it if the reviewer could kindly point to any concern that may have been left unaddressed. We believe that by addressing the suggested changes from the reviewer, the current draft will be a good addition to the literature on equivariant networks.

---

### Official Review · Reviewer_YzuG · 2023-11-01

**Soundness:** 3 good
**Presentation:** 2 fair
**Contribution:** 3 good
**Rating:** 5
**Confidence:** 3

**Summary:**

The authors propose a simple neural network architecture that is equivariant to arbitrary matrix groups. The architecture makes use of scalar, vector, and higher-order tensor representations. Operations consist of transforming between these representations, and linear mixing of representations of the same type. The authors conceptually compare their work to EMLP (Finzi et al.) and argue that their architecture is simpler, more efficient, and scales better to larger datasets and data types. Experiments are provided for:
- Symmetries such as O(5), O(1, 3), and O(3).
- Image classification under rotations.
- Predicting n-body dynamics with GNNs.
- Solving PDEs.

**Strengths:**

* The simplicity and generality of the architecture is appealing. I think this work should be seen as a generalization of the vector neurons architecture. Perhaps a more suitable name for the architecture would have been tensor neurons. To my knowledge, the vector neurons framework had not previously been generalized so the work is novel to me.
* The experiments are varied and touch on several different application areas of ML.

**Weaknesses:**

* As I've mentioned, I see this work as a generalization of the vector neurons architecture. The generalization is, however, very straightforward and requires no technical work. Even the proof of equivariance carries over almost directly.
* Even though the architecture is quite general, the authors use ad-hoc methods for designing the architectures in the experiments and rely on previously obtained models for symmetric data such as convolution for images and message passing for graphs. It might be worth mentioning that, in principle, all models in the paper could have followed the exact same blueprint (with a possible drop in performance). For example, the translation symmetry of images and the permutation symmetry of graphs could all have been handled under the GRepsNet architecture. The paper could be more clear in describing which sub-symmetries are being handled by their model.
* The work shouldn't be compared to EMLP (Finzi et al.). As I've mentioned, comparing to vector neurons is much more appropriate. What EMLP does is that it computes *all* equivariant bases, which is much more challenging than providing *some* equivariant bases, as in this work. EMLP is a valid baseline of comparison in experiments, but the goal of that paper is different from this one.
* For the image experiments there should be a comparison with group convolutional networks (Cohen & Welling).

**Questions:**

- Are there non-linearities being applied to T_0 representations (scalars)? The T_i (i > 0) reps don't have nonlinearities so I presume that nonlinearity must come in from being mixed with T_0 reps but the paper only says that this is done for better mixing. Also, I think this is how different indices of the T_i tensors are communicating information, i.e., without this mixing, each index of the T_i tensors would be processed entirely separately. It's not clearly explained in the paper.
- Does the model provide universal approximation guarantees?

---

> ### Author Response · Authors · 2023-11-22
> **Response to reviewer YzuG**
>
> We thank the reviewer for their detailed comments and suggestions. We address the concerns raised below.
>
> **Reviewer:** As I've mentioned, I see this work as a generalization of the vector neurons architecture. The generalization is, however, very straightforward and requires no technical work. Even the proof of equivariance carries over almost directly.
>
> **Response:** We agree with the reviewer that this work can be seen as a generalization of the vector neuron architecture as mentioned in Sec. 4.2.
>
> We agree that the generalization is simple in theory. But, it is a crucial generalization that makes GRepsNet work with any general data type and makes it competitive to more sophisticated architecture such as EMLPs. Further, we apply our architecture in a wide variety of domains achieving competitive performs with SOTA models across many of them, e.g. with GCNNs and E(2)-CNNs for image classification (now added in Tab. 7 and 8), etc.
>
> Hence, we believe that the simplicity of our model makes it appealing for use in a wide range of domains.
>
> **Reviewer:** Even though the architecture is quite general, the authors use ad-hoc methods for designing the architectures in the experiments ... The paper could be more clear in describing which sub-symmetries are being handled by their model.
>
> **Response:** We have now updated Sec. 4.3 to give an overview of our design principle. The main design choices made are simply the input representations types, i.e. regular vs. non-regular group representations. We hope this gives a more unified view of the design choices made in our work.
>
> We have now also added the groups of symmetries considered for each task in Sec. 4.3, e.g., group of 90-degree rotations for pde solving with FNOs, O(3) for N-body dynamics, just like in previous works G-FNOs and EGNNs, respectively. We have only considered these subsymmetries so as to have a fair comparison with the baselines, e.g. GCNNs use group convolutions only to add the symmetries corresponding to the 90-degree rotations (C4 group) on top of CNNs that are already translation equivariant. Hence, when comparing with GCNNs, we simply use the C4 group symmetry for GRepsCNNs on top of CNNs that are already translation equivariant.
>
> **Reviewer:** The work shouldn't be compared to EMLP (Finzi et al.). As I've mentioned, comparing to vector neurons is much more appropriate. What EMLP does is that it computes all equivariant bases, which is much more challenging than providing some equivariant bases, as in this work. EMLP is a valid baseline of comparison in experiments, but the goal of that paper is different from this one.
>
> **Response:** We agree with the reviewer. Our architecture is not primarily meant to be compared with EMLPs. Instead, our goal is to design a general purpose equivariant network inspired by EMLPs, but designed to be much simpler, faster, and competitive in performance. This is also justified by our experiments, where comparison with EMLPs is only one subsection. Whereas a majority of the experiments focus on larger datasets (that EMLPs cannot be used with due to its complexity), and leverage the benefits of equivariance across many domains, often performing competitively with the SOTA models. Since VNs are already a special case of our design, as described in Sec. 4.2, we do not compare them in experiments.
>
> **Reviewer:** For the image experiments there should be a comparison with group convolutional networks (Cohen & Welling).
>
> **Response:** We have now added baseline comparisons for image classification with GCNNs and E(2)CNNs. We find that not only is GRepsCNNs are competitive with both baselines, but when we use second-order tensors in these SOTA models, they outperform their original first-order counterparts. That it, when we introduce second-order features in GCNNs and E(2)CNNs, let's call them T_2-GCNNs and T_2-E(2)CNNs, respectively, then they outperform the original GCNNs and E(2)CNNs in image classification. This is similar to our finding with second order equituning in Sec. 5.5.
>
> **Reviewer:** Are there non-linearities being applied to T_0 representations (scalars)? ...
>
> **Response:** Yes, we do add non-linearities in the T_0 layer. We have now written this explicitly in Sec. 4.1.
>
> **Reviewer:** Does the model provide universal approximation guarantees?
>
> **Response:** Yes, we have now added a simple constructive universality proof of GRepsNet for the O(d), O(1, d) groups in Sec. G of the appendix.

---

### Official Review · Reviewer_FzvF · 2023-11-03

**Soundness:** 1 poor
**Presentation:** 1 poor
**Contribution:** 1 poor
**Rating:** 1
**Confidence:** 5

**Summary:**

The paper considers the problem of building transformation-equivariant and invariant neural networks. The authors' main focus is to create neural networks that are scalable and generally applicable. They propose GRepsNet - a class of neural networks that take tensor features as inputs and process them with T-layers.

The structure of the paper does not allow a reader to understand the method clearly. The way the paper presents its ideas is not sufficient for implementation. Moreover, while the paper contains a lot of theory and very general explanations, it lacks details on the method itself. The story of the paper is not coherent.

It is very complicated to understand the exact implementation, the exact contribution, and the specific details of the performed research from the current version of the paper. It requires a major revision to meet the high standards of the ICLR conference.

**Strengths:**

The paper has some strengths by themselves, which, however, do not make the paper coherent. Among the strengths I can mention a thorough explanation of some of the concepts such as group representations.

**Weaknesses:**

The main weaknesses of the paper are that it contains a lot of technical details on topics that may be familiar to the reader, and it does not provide enough information to form a strong understanding of the proposed method.

- The structure of the paper should be changed in order to improve its readability. The standard "Intro, Related Work, Theory, Experiments, Concluiosn" can serve as an example.
- The entire Section 2, "Background," seems like a repetition of what a reader may already know from references. Thus, it's not necessary to present it in such explicit detail. It is a mix of "related work" and "prelimiries for the theory section", but it doesn't serve any of these roles.
- It's not clear what type of neural network is being built. What is the main contribution of the paper? Is it proposing the use of different features, or is it introducing a new type of neural network? Is it a marginal improvement over EMLP? Is it the next interation of Villar et al? Is it an absolutely new approach for building equivariant models? If so, why don't you compare it against SOTA competitors?
- The choice of datasets and models in the experiments is not clear. Why was MLP Mixer chosen instead of a group equivariant CNN?

**Questions:**

- Can you compare your method to a group equivariant neural network with a similar number of parameters? I can suggest to use a CNN from *Weiler M., Cesa G. General e (2)-equivariant steerable cnns NeurIPS 2019*
- Page 4, "Process converted tensors", you write that "we simply pass it through a linear neural network with no point-wise non-linearities or bias terms to ensure that the output is equivariant". What is a linear NN here? Is it a simple matrix multiplication?

---

> ### Author Response · Authors · 2023-11-22
> **Response to reviewer FzvF**
>
> We thank the reviewer for their detailed feedback. We address the concerns raised by the reviewer below.
>
> **Reviewer:** "... it does not provide enough information to form a strong understanding of the proposed method ...", "The structure of the paper should be changed in order to improve its readability. The standard "Intro, Related Work, Theory, Experiments, Concluiosn" can serve as an example."
>
> **Response:** We have added a "Related work" section after the "Intro" and moved the detailed discussion with EMLPs and Universal Scalars to the appendix. We believe this helps with better readability and provides a better comparison with various works in literature.
>
> We have now added a paragraph in Sec. 4.3 to better explain how our general construction provided in Sec. 4.1 is used to design the architectures for various applications. Further, we have added more details on the exact networks designed for our experiments in Sec. D in the appendix. We hope these changes help clarify our method better.
>
> **Reviewer:** It's not clear what type of neural network is being built. What is the main contribution of the paper? Is it proposing the use of different features, or is it introducing a new type of neural network? Is it a marginal improvement over EMLP? Is it the next interation of Villar et al? Is it an absolutely new approach for building equivariant models? If so, why don't you compare it against SOTA competitors?
>
> **Response:** The main goal of our work is to provide a simple, general, and efficient algorithm to construct equivariant architectures: GRepsNet is easy to construct, works across a wide range of domains, and provide competitive performance to SOTA equivariant models.
>
> It is not a marginal improvement over EMLP, rather, it provides a much simpler architecture for the same type of equivariance. It is related to Villar et al., but the emphasis is to design more practical networks than Villar et al. Note that Villar et al. mostly focus on construction of universal architectures, but only experiment with synthetic data since it is not trivial to apply it on larger datasets of more practical importance such as image datasets. Hence, we design equivariant networks that are computational efficient and hence of practical importance.
>
> The key takeway from our work is the importance of feature representations in equivariant networks. We show that by simply using suitable feature representations, we can obtain competitive performance with SOTA models that use much complicated architectures and can be computationally expensive.
>
> Moreover, the use of higher order tensors in equivariant finetuning (Tab. 5), GCNNs (now added Tab. 7) and E(2)-CNNs (now added Tab. 8) outperform their traditional first order representation based architectures. Hence, again, emphasizing the importance of group representation in constructing group equivariant networks.
>
> **Reviewer:** The choice of datasets and models in the experiments is not clear. Why was MLP Mixer chosen instead of a group equivariant CNN?
>
> **Response:** We have now added comparison of GRepsCNNs with GCNNs in Tab. 7 finding that GRepsCNN perform competitively with GCNNs. Moreover, we extend GCNNs, which use first order regular representations, to second order representations. By doing so, we outperform the original GCNNs of Cohen et al. in image classification task. This shows that using higher order tensor representations is a powerful tool in equivariant representation learning.
>
> **Reviewer:** Can you compare your method to a group equivariant neural network with a similar number of parameters ... CNN from Weiler M., Cesa G. General e (2)-equivariant steerable cnns NeurIPS 2019
>
> **Response:** We have now added comparison of GRepsCNNs with E(2)CNNs in Tab. 8 finding that GRepsCNN perform competitively with E(2)CNNs. Moreover, we extend E(2)CNNs, which use first order regular representations, to second order representations.
>
> By doing so, we outperform the original E(2)CNNs of Weiler et al. in image classification task. This shows that using higher order tensor representations is a powerful tool in equivariant representation learning.
>
> **Reviewer:** Page 4, "Process converted tensors", you write that "we simply pass it through a linear neural network with no point-wise non-linearities or bias terms to ensure that the output is equivariant". What is a linear NN here? Is it a simple matrix multiplication?
>
> **Response:**  By linear NN, we simply mean a sequence of matrix multiplications.

---

### Official Review · Reviewer_rnny · 2023-11-05

**Soundness:** 3 good
**Presentation:** 3 good
**Contribution:** 4 excellent
**Rating:** 8
**Confidence:** 4

**Summary:**

This paper presents a general framework for building equivariant networks for any matrix group. The proposed method is far more scalable than prior work like EMLPs (Finzi et al.). It is also very general and can be combined with advanced architectures including MLP mixers, message-passing neural networks, Fourier neural operators, etc for various applications. The authors also introduce using higher-order tensor representations and show this improves performance.

**Strengths:**

- This work could have significant impact. The proposed framework is very general, able to construct equivariant networks for any matrix group. It can also be combined with many advanced deep learning architectures across diverse applications. Additionally, the method scales to large problems. There is strong potential for numerous applications of this work.
- The authors provide a unified view of previous work such as vector neurons, harmonic networks and equitune. Their framework also enables incorporating higher-order tensor representations. In this work, they already verified that second-order tensors are useful as equivariant features.

**Weaknesses:**

- The authors' method provides a very general framework that can process tensor representations $T_i$ of arbitrary order. However, in practice, most experiments only use $T_0$ and $T_1$, with one additional experiment testing $T_2$. While this paper still makes an important contribution, there is a disconnect between the theory and experiments - the importance of higher-order ($i>2$) tensor representations is not empirically demonstrated. I'm simply noting that the full generality of the framework is not validated, even though the core ideas represent an advance.

**Questions:**

- Do the authors know if their model is universal, in the sense that any equivariant function with respect to the matrix groups can be well approximated by the model given a suitable parameter set? Are there any subsets of equivariant functions that cannot be represented? With that said, the paper still makes significant contributions by proposing a general framework for constructing equivariant networks and demonstrating strong empirical performance. Further analysis of the model's theoretical expressivity would be an interesting area for future work.

**Details Of Ethics Concerns:**

I have no ethical concerns.

---

> ### Author Response · Authors · 2023-11-22
> **Response to reviewer rnny**
>
> We thank the reviewer for their positive comments and suggestions.
>
> **Weakness:** We agree that our framework is very general and can handle any tensors of higher orders. However, all of our experiments only required tensors up to order 2, hence, we restrict ourselves with second-ordered tensors. We leave applications to higher-ordered tensors for future work.
>
> **Question:** We thank the reviewer for pointing us to the universality of our method. We now add a simple constructive proof showing that our model is indeed a universal approximator of equivariant vector functions and invariant scalar functions for O(d) and O(1, d) groups in Sec. G in the appendix.

---

### Author Response · Authors · 2023-11-22
**Summary of responses to all reviewers**

We thank the reviewers for their detailed feedback. Here, we summarize our responses to the comments and suggestions from reviewers and the changes we made to the paper.

**Main contribution and clarification of the proposed method:** As requested by reviewers FzvF, YzuG, we clarify the main contribution and clarify our architecture design.
The main goal of our work is to provide a simple, general, and efficient algorithm to construct equivariant architectures: GRepsNet is easy to construct, works across a wide range of domains, and provide competitive performance to SOTA equivariant models.

Additionally, we provide a unified overview of our construction algorithm in Sec. D of the paper to help clarify the construction algorithm.

**Comparison to baselines:** Based on the requests by reviewers FzvF, YzuG, we have now added baseline comparisons for image classification with GCNNs and E(2)CNNs. We find that not only is GRepsCNNs are competitive with both baselines, but when we use second order tensors in these SOTA models, they outperform their original first order counterparts. That it, when we introduce second order features in GCNNs and E(2)CNNs, let's call them T_2-GCNNs and T_2-E(2)CNNs, respectively, then they outperform the original GCNNs and E(2)CNNs in image classification. This is similar to our finding with second order equituning in Sec. 5.5.


**Universality/expressivity of our architecture:** Based on concerns raised by reviewers rnny, YzuG, and 6e6R about the expressivity/universality of our architecture, we now add a simple constructive proof showing that our model is indeed universal approximator of equivariant vector functions and invariant scalar functions for O(d) and O(1, d) groups in Sec. G.

---

### Meta-Review · Area_Chair_6hJ1 · 2023-12-07

**Metareview:**

The authors introduce an algorithm for building equivariant neural networks to arbitrary matrix groups which uses tensor representations for the hidden feature representations.   They apply the method to several different applications including image classification, N-body dynamics prediction, and solving PDEs.

Many reviewers found the method to indeed be simple, general, and efficient and praised the varied experimental domains.   However, there were widely shared concerns. The clarify of the description of the method was found lacking. There are also significant issues with the experimental evaluation.  Although the method is defined generally, experiments were limited orthogonal groups and order 2 tensors.  Moreover, the comparison with baselines is not sufficient to convince that the method provides an advantage.  Reviewers noted missing baselines in image classification and the n-body problem.  Although the authors added experiments with E(2)-CNN the experiments were performed under different settings than those established in the original paper (also an issue with comparison to Finzi et. al) and do not provide enough confidence in the empirical performance of the method. Rigorous benchmark comparisons would strengthen the paper.

**Justification For Why Not Higher Score:**

- clarity
- adequately rigorous comparison to correct baselines demonstrating architecture across different groups and tensor orders.

**Justification For Why Not Lower Score:**

N/A

---

### Decision · Program_Chairs · 2024-01-16

Reject